# Obesity and risk of female reproductive conditions: A Mendelian randomisation study

**Samvida S. Venkatesh**[1,2]*, Teresa Ferreira[1,3], Stefania Benonisdottir[1],
**Nilufer Rahmioglu**[2,3], Christian M. Becker[3], Ingrid Granne[3], Krina T. Zondervan[2,3],
**Michael V. Holmes**[4,5], Cecilia M. Lindgren[1,2,3,4,6], Laura B. L. Wittemans[1,3]*

1 Big Data Institute, Li Ka Shing Centre for Health Information and Discovery, University of Oxford, Oxford, United Kingdom, 2 Wellcome Centre for Human Genetics, Nuffield Department of Medicine, University of Oxford, Oxford, United Kingdom, 3 Nuffield Department of Women's and Reproductive Health, Medical Sciences Division, University of Oxford, Oxford, United Kingdom, 4 Nuffield Department of Population Health, University of Oxford, Oxford, United Kingdom, 5 Medical Research Council Population Health Research Unit, University of Oxford, Oxford, United Kingdom, 6 Broad Institute of MIT and Harvard, Cambridge, Massachusetts, United States of America

* samvida@well.ox.ac.uk (SSV); laura.wittemans@wrh.ox.ac.uk (LBLW)

## Abstract

### Background

Obesity is observationally associated with altered risk of many female reproductive conditions. These include polycystic ovary syndrome (PCOS), abnormal uterine bleeding, endometriosis, infertility, and pregnancy-related disorders. However, the roles and mechanisms of obesity in the aetiology of reproductive disorders remain unclear. Thus, we aimed to estimate observational and genetically predicted causal associations between obesity, metabolic hormones, and female reproductive disorders.

### Methods and findings

Logistic regression, generalised additive models, and Mendelian randomisation (MR) (2-sample, non-linear, and multivariable) were applied to obesity and reproductive disease data on up to 257,193 women of European ancestry in UK Biobank and publicly available genome-wide association studies (GWASs). Body mass index (BMI), waist-to-hip ratio (WHR), and WHR adjusted for BMI were observationally (odds ratios [ORs] = 1.02–1.87 per 1-SD increase in obesity trait) and genetically (ORs = 1.06–2.09) associated with uterine fibroids (UF), PCOS, heavy menstrual bleeding (HMB), and pre-eclampsia. Genetically predicted visceral adipose tissue (VAT) mass was associated with the development of HMB (OR [95% CI] per 1-kg increase in predicted VAT mass = 1.32 [1.06–1.64], $P = 0.0130$), PCOS (OR [95% CI] = 1.15 [1.08–1.23], $P = 3.24 \times 10^{-05}$), and pre-eclampsia (OR [95% CI] = 3.08 [1.98–4.79], $P = 6.65 \times 10^{-07}$). Increased waist circumference posed a higher genetic risk (ORs = 1.16–1.93) for the development of these disorders and UF than did increased hip circumference (ORs = 1.06–1.10). Leptin, fasting insulin, and insulin resistance each mediated between 20% and 50% of the total genetically predicted association of obesity with pre-eclampsia. Reproductive conditions clustered based on shared genetic components of their aetiological relationships with obesity. This study was limited in power by the

**Data Availability Statement:** All scripts used in analyses are deposited at https://github.com/lindgrengroup/obesity_femrepr_MR. Data are

available from UK Biobank (https://www.ukbiobank.ac.uk/) in accordance with their published data access procedures. Summary data for the FinnGen GWAS projects are available from their results portal (https://www.finngen.fi/en/access_results). Other publicly available GWAS summary data are available from the GWAS Catalog (https://www.ebi.ac.uk/gwas/). All other relevant data are within the manuscript and its Supporting Information files.

**Funding:** S.S.V. is supported by the Rhodes Scholarships (https://www.rhodeshouse.ox.ac.uk/), Clarendon Fund (http://www.ox.ac.uk/clarendon/about), and the Medical Sciences Doctoral Training Centre (https://www.medsci.ox.ac.uk/) at the University of Oxford. S.B. is supported by the Li Ka Shing Foundation. M.V.H. works in a unit that receives funding from the UK Medical Research Council and is supported by a British Heart Foundation Intermediate Clinical Research Fellowship (FS/18/23/33512) and the National Institute for Health Research Oxford Biomedical Research Centre. C.M.L. is supported by the Li Ka Shing Foundation, NIHR Oxford Biomedical Research Centre, Oxford, NIH (1P50HD104224-01), Gates Foundation (INV-024200), and a Wellcome Trust Investigator Award (221782/Z/20/Z). L.B.L.W. is supported by the Wellcome Trust (221651/Z/20/Z). The research was supported by the Wellcome Trust Core Award Grant Number 203141/Z/16/Z with additional support from the NIHR Oxford BRC. The funders had no role in study design, data collection and analysis, decision to publish, or preparation of the manuscript.

**Competing interests:** I have read the journal's policy and the authors of this manuscript have the following competing interests: C.M.B. reports grants from Bayer AG, AbbVie Inc, Volition Rx, MDNA Life Sciences, Roche Diagnostics Inc., and consultancy for Myovant. He is a member of the independent data monitoring board at ObsEva; I.G. reports grants from Bayer AG; K.T.Z. reports grants from Bayer AG, AbbVie Inc, Volition Rx, MDNA Life Sciences, Roche Diagnostics Inc, and non-financial scientific collaboration with Population Diagnostics Ltd, outside the submitted work; M.V.H. has consulted for Boehringer Ingelheim, and in adherence to the University of Oxford's Clinical Trial Service Unit & Epidemiological Studies Unit (CSTU) staff policy, did not accept personal honoraria or other payments from pharmaceutical companies; C.M.L. reports grants from Bayer AG and Novo Nordisk and has a partner who works at Vertex; no other relationships or activities that could appear to have influenced the submitted

low prevalence of female reproductive conditions among women in the UK Biobank, with little information on pre-diagnostic anthropometric traits, and by the susceptibility of MR estimates to genetic pleiotropy.

## Conclusions

We found that common indices of overall and central obesity were associated with increased risks of reproductive disorders to heterogenous extents in a systematic, large-scale genetics-based analysis of the aetiological relationships between obesity and female reproductive conditions. Our results suggest the utility of exploring the mechanisms mediating the causal associations of overweight and obesity with gynaecological health to identify targets for disease prevention and treatment.

## Author summary

### Why was this study done?

- Female reproductive disorders are common, yet relatively understudied, conditions with a large impact on women's health and overall well-being.

- Obesity has previously been associated with the risk of developing female reproductive conditions, but estimates may be biased by weight gain caused by the disease or its treatment, as well as confounding environmental and lifestyle factors.

### What did the researchers do and find?

- In one of the largest publicly available datasets on human health, which includes up to 250,000 women, we saw positive associations between obesity and a range of female reproductive disorders, including uterine fibroids, polycystic ovary syndrome, heavy menstrual bleeding, and pre-eclampsia.

- We found that inherited genetic variation that is associated with obesity is also associated with female reproductive disorders, but the strength of these associations differs by type of obesity and reproductive condition. As genetic variants are randomly assigned at birth, this is a method to estimate the effect of obesity on reproductive conditions unbiased by environmental and lifestyle factors or reverse causation.

- Hormones such as leptin, which is secreted by fat cells, and insulin were found to mediate the genetic association of obesity with pre-eclampsia.

### What do these findings mean?

- Genetics-based investigations such as these provide support for the role of obesity in increasing the risk of reproductive conditions, reinforcing the need to address rising obesity rates in the population.

work. All disclosed competing interests are outside the submitted work.

**Abbreviations:** AIC, Akaike's information criterion; BMI, body mass index; FDR, false discovery rate; GAM, generalised additive model; GIANT, Genetic Investigation of ANthropometric Traits; GWAS, genome-wide association study; HC, hip circumference; HMB, heavy menstrual bleeding; ISI, Stumvoll Insulin Sensitivity Index; IVW, inverse-variance-weighted; MR, Mendelian randomisation; OR, odds ratio; PCOS, polycystic ovary syndrome; UF, uterine fibroid; UKBB, UK Biobank; UMAP, Uniform Manifold Approximation and Projection; VAT, visceral adipose tissue; WC, waist circumference; WHR, waist-to-hip ratio; WHRadjBMI, waist-to-hip ratio adjusted for body mass index.

- Increased obesity and insulin resistance are potentially modifiable risk factors, and addressing these risk factors could help mitigate or treat female reproductive disorders, but further research must confirm that their manipulation influences risk of disease.

## Introduction

Obesity is commonly understood as the excess accumulation of body fat, which leads to increased health risks. In women, increased body mass index (BMI) is associated with increased prevalence of gynaecological conditions, including excessive and abnormal menstrual bleeding [1,2], endometriosis and uterine fibroids (UF) [3,4], polycystic ovary syndrome (PCOS) [5,6], complications of pregnancy such as pre-eclampsia and eclampsia [7], miscarriage [8,9], and infertility [10,11]. These are often non-linear and heterogeneous relationships. While the risks of anovulatory infertility and recurrent miscarriages are highest in obese women, underweight women also have increased risk of infertility [9,12]. The association of BMI with endometriosis varies by disease severity, as women with advanced-stage endometriosis have lower BMI than those with minimal disease, and the inverse BMI–endometriosis association is stronger in women with infertility [13,14]. Finally, although the severity of PCOS and menstrual disorders increases with overall obesity, women presenting with these conditions are more likely to store fat in the abdominal region, regardless of their BMI [2,5].

Previous estimates of the associations of obesity with reproductive conditions have primarily been based on observational study designs including case–control studies [8,15,16] and cross-sectional studies in randomly selected women [1], or cross-sectional studies conducted only in women with obesity [2]. While the direction of effect is largely consistent across studies, the heterogeneity in selection of cases, controls, and populations observed in these studies is reflected in heterogeneity in the effect estimates. Further, observational epidemiological studies are limited in assessing causality, due to confounding and reverse causation. The Mendelian randomisation (MR) framework is a genetics-based instrumental variable approach that relies on the random and fixed assignment of genetic variants at conception to estimate the causal effect size of genetically predicted exposures on an outcome. MR has previously indicated causal associations of genetically predicted BMI with some subtypes of ovarian cancer (OR = 1.29 per 5 units of BMI) [17], endometrial cancer (OR = 2.06 per 5 units of BMI) (16), and PCOS (OR = 4.89 per 1 SD higher BMI) [18]. However, the aetiological role of obesity and body fat distribution in many other female reproductive diseases has not been reported. It is especially relevant to investigate the effects of fat distribution, as there are intricate metabolic and endocrine links between adipose tissue and female reproductive organs. Yet, causal investigations of such relationships are lacking.

Leptin, which is a hormone secreted by adipocytes and elevated in individuals with obesity, is increased in women with endometriosis, UF, and hypertensive disorders of pregnancy, even when analyses are adjusted for BMI [7,19–22]. Obesity-induced insulin resistance additionally increases the risk and severity of PCOS and pre-eclampsia by dysregulating steroid hormone and metabolic pathways [5,23,24]. The dysregulation of sex hormones, including oestrogen and testosterone, is likely to play a role in the obesity-driven development of female reproductive disorders due to its close associations with body fat [23,25]. Yet, to the best of our

knowledge, the causal impact of these factors in mediating the relationships between obesity and gynaecological diseases has not been detailed.

Here, we apply logistic regression, generalised additive models (GAMs), and 2-sample, non-linear, and multivariable MR to dissect the relationships of overall obesity and body fat distribution with a range of female reproductive disorders, and to investigate the mediating role of metabolic factors including leptin and insulin.

## Methods

### Observational associations in UK Biobank

UK Biobank (UKBB) is a prospective UK-based cohort study with approximately 500,000 participants aged 40–69 years at recruitment on whom a range of medical, environmental, and genetic information is collected [26]. We included 257,193 individuals self-identifying as females of white ancestry in UKBB in our analyses. Baseline measurements of BMI (total body weight [kg]/standing height squared [$m^2$]) and waist-to-hip ratio (WHR) (waist circumference [WC] [cm]/hip circumference [HC] [cm]), and WHR adjusted for BMI (WHRadjBMI) were used to estimate general obesity (BMI) and central obesity (WHR and WHRadjBMI). In response to peer review comments, comparative body size at age 10 years (as self-reported in a questionnaire with the options 'thinner', 'plumper', or 'about average') was used to estimate adiposity at an earlier time point, i.e., before diagnosis of most reproductive disorders. Cases of reproductive conditions were identified based on ICD-9 and ICD-10 primary and secondary diagnoses from hospital inpatient data, self-reported illness codes, and primary care records (Table 1). We fitted logistic regression models to estimate the associations of BMI, WHR, and WHRadjBMI with prevalence of endometriosis (7,703 cases, 249,490 controls); heavy menstrual bleeding (HMB) (17,229 cases, 239,964 controls); infertility (2,194 cases, 254,999 controls); self-reported stillbirth, spontaneous miscarriage, or termination (81,102 cases, 176,091 controls); PCOS (746 cases, 256,447 controls); pre-eclampsia (2,242 cases, 254,951 controls); and UF (19,192 cases, 238,001 controls). Case definitions for pre-eclampsia included eclampsia cases to capture cases in which the former may have developed into the latter. For each disease, individuals not included in the case group were used as controls. BMI, WHR, and WHRadjBMI were adjusted for age, age squared, assessment centre, and smoking status. The residuals were rank-based inverse normally transformed. Multiple testing correction for 21 tests (3 exposures × 7 outcomes) was applied using the false discovery rate (FDR) to evaluate statistical significance while minimising false negatives [27,28].

We also tested associations without adjustment for smoking status, as it has previously been suggested that higher BMI increases risk of smoking [29] and adjustment for both could therefore induce collider bias. Adjustment for menopause status was not performed as up to 42% of women with reproductive disorders in UKBB report being unsure of their menopause status, as compared to 16% of women who do not have a recorded history or presence of a reproductive condition (Table 1).

To evaluate the presence of non-linear observational associations between obesity and each reproductive trait, fractional polynomial regression following the closed test procedure was performed using the mfp v1.5.2 R package [30]. This algorithm tests for the presence of an overall association, determines the likelihood of non-linearity, and selects the best-fitting fractional polynomial function. We also fitted GAMs to the same data, allowing for smoothing of the obesity trait with splines, using the mgcv 1.8–31 R package [31]. These models allow a greater degree of flexibility in modelling curves that cannot be represented by polynomials of the *n*th degree; however, they are also more complex and thus less immediately interpretable [32]. All models were adjusted for age, age squared, assessment centre, and smoking status.

**Table 1. Summary of female reproductive disorders in UK Biobank.**

| Diagnosis | Sample size (prevalence) | Mean age (years) (SD) | Mean BMI (kg/m$^2$) (SD) | Mean WHR (SD) | Smoking prevalence | Menopause status | Phenotype definition |
|---|---|---|---|---|---|---|---|
| None | 148,493 (57.8%) | 57.22 (7.85) | 26.89 (5.05) | 0.82 (0.07) | Never—61% Previous—31% Current—8% | Yes—66% No—21% Not sure—13% | |
| Endometriosis | 7,703 (2.99%) | 53.46 (7.89) | 27.57 (5.34) | 0.82 (0.07) | Never—59% Previous—30% Current—10% | Yes—34% No—23% Not sure—42% | ICD-10 primary and secondary diagnosis and cause of death: N80[0–9] ICD-9 primary and secondary diagnosis: 617[0–9] Self-reported non-cancer illness code: 1402 endometriosis Primary care records |
| Heavy menstrual bleeding | 17,229 (6.70%) | 52.52 (7.08) | 27.86 (5.61) | 0.82 (0.07) | Never—58% Previous—30% Current—11% | Yes—38% No—32% Not sure—29% | ICD-10 primary and secondary diagnosis and cause of death: N92[0–6] ICD-9 primary and secondary diagnosis: 626[2\|3\|4\|5\|6\|8\|9] Primary care records |
| Infertility | 2,194 (0.85%) | 48.90 (6.68) | 26.09 (4.85) | 0.80 (0.07) | Never—63% Previous—27% Current—9% | Yes—31% No—54% Not sure—14% | ICD-10 primary and secondary diagnosis and cause of death: N97[0\|1\|2\|3\|4\|8\|9] ICD-9 primary and secondary diagnosis: 628[0\|1\|2\|3\|4\|8\|9] Self-reported non-cancer illness code: 1403 (female infertility) Primary care records |
| Stillbirth, spontaneous miscarriage, or termination | 81,102 (31.5%) | 55.84 (8.04) | 27.10 (5.22) | 0.82 (0.07) | Never—53% Previous—35 Current—12% | Yes—58% No—25% Not sure—16% | ICD-10 primary and secondary diagnosis and cause of death: O03[0–9] ICD-9 primary and secondary diagnosis: 634[0–9] Self-reported non-cancer illness code: 1559 (miscarriage) Self-reported: 'Ever had stillbirth, spontaneous miscarriage, or termination' Primary care records |
| Polycystic ovary syndrome | 746 (0.29%) | 47.87 (6.83) | 30.45 (7.57) | 0.83 (0.08) | Never—62% Previous—28% Current—9% | Yes—19% No—60% Not sure—21% | ICD-10 primary and secondary diagnosis and cause of death: E282 ICD-9 primary and secondary diagnosis: 2564 Self-reported non-cancer illness code: 1350 (polycystic ovaries/polycystic ovarian syndrome) Primary care records |
| Pre-eclampsia (or eclampsia) | 2,242 (0.87%) | 53.69 (8.66) | 28.00 (5.57) | 0.82 (0.07) | Never—65% Previous—28% Current—7% | Yes—48% No—38% Not sure—14% | ICD-10 primary and secondary diagnosis and cause of death: O14[0\|1\|2\|9], O15[0\|1\|2\|9] ICD-9 primary and secondary diagnosis: 642[4\|5\|6\|7] Self-reported non-cancer illness code: 1073 (gestational hypertension/pre-eclampsia) Primary care records |
| Uterine fibroids | 19,192 (7.46%) | 56.86 (7.51) | 27.63 (5.26) | 0.82 (0.07) | Never—60% Previous—32% Current—7% | Yes—46% No—17% Not sure—37% | ICD-10 primary and secondary diagnosis and cause of death: O14[0\|1\|2\|9], D25[0\|1\|2\|9] ICD-9 primary and secondary diagnosis: 218[9] Self-reported non-cancer illness code: 1351/1352 (uterine fibroids/uterine polyps) Primary care records |

BMI, body mass index; ICD-9/10, International Classification of Diseases–Revision 9/10; SD, standard deviation; WHR, waist-to-hip ratio.

Model fits were compared with Akaike's information criterion (AIC), with lower AIC indicating better fit [33].

As a data-driven investigation, the analyses upon which this paper is based were planned and designed for each of the sections independently; no overall prospective analysis plan was followed. Analyses dependent upon results from other sections, such as the mediation analysis dependent on associations from 2-sample MR, are noted as such within each section. This study is reported as per the Strengthening the Reporting of Observational Studies in Epidemiology Using Mendelian Randomization (STROBE-MR) guideline (S1 Checklist).

## Two-sample MR

Genetic instruments for BMI, WHR, and WHRadjBMI were selected based on the sentinel variants at genome-wide significant loci ($P < 5 \times 10^{-9}$ to account for denser imputation data) reported in the largest publicly available European ancestry genome-wide association studies (GWASs), which are meta-analyses of the Genetic Investigation of ANthropometric Traits (GIANT) and UKBB (maximum $N$ = 806,801 individuals) [34]. Similarly, genetic instruments for predicted visceral adipose tissue (VAT) mass ($N$ = 325,153 individuals) [35], WC ($N$ = 462,166 individuals), HC ($N$ = 462,117 individuals) [36], and waist-specific and hip-specific WHR (i.e., WHR instruments with specific effects on WC but not HC and vice versa; $N$ = 18,330 individuals) [37] were selected based on the largest publicly available GWASs.

Three instrument weighting strategies were considered where sex-stratified GWAS results were available: (i) SNPs from combined-sex GWASs with combined-sex weights (effect sizes), (ii) combined-sex SNPs with female-specific weights, and (iii) female-specific SNPs with female-specific weights. The method of female-specific SNPs with female-specific weights produced the strongest instruments as evaluated by $F$-statistics, i.e., mean $\beta^2/\sigma^2$ over all SNPs in the instrument, and was thus chosen for analysis (S1 Table). Additionally, due to concerns of ascertainment bias in UKBB [38,39], sensitivity analyses with combined-sex instruments (combined-sex SNPs with combined-sex weights) were also performed. In response to peer review comments, an additional sensitivity analysis with SNPs from previous GIANT releases for BMI [40] and WHR and WHRadjBMI [41] that do not include UKBB participants was also performed to alleviate potential ascertainment bias. Finally, to alleviate concerns of collider bias in the WHRadjBMI GWAS [42], we constructed a joint WHR and BMI instrument to perform multivariable MR.

Associations of the genetic instruments for obesity traits with female reproductive diseases were obtained by performing a fixed-effect inverse-variance-weighted (IVW) meta-analysis of publicly available GWAS summary statistics from 2 large biobank projects—FinnGen and UKBB [43]. The meta-analysis was performed using METAL [44] by matching the relevant ICD codes (S2 Table) for the following traits: infertility (4,996 cases, 421,223 controls), preeclampsia (2,711 cases, 480,373 controls), and UF (21,835 cases, 456,551 controls). For endometriosis, summary statistics were obtained by request from a recent European ancestry GWAS [45] and meta-analysed as above with publicly available FinnGen and UKBB summary statistics (12,210 cases, 450,183 controls). For HMB (9,813 cases, 210,946 controls), sporadic miscarriage (i.e., 1–2 miscarriages; 50,060 cases, 174,109 controls), and multiple consecutive miscarriage (i.e., ≥3 consecutive miscarriages; 750 cases, 150,215 controls), publicly available summary statistics were obtained from recent European ancestry GWASs that include UKBB individuals [3,46]. For PCOS, estimates were based on a fixed-effect IVW meta-analysis of published GWAS summary statistics [47], publicly available GWAS results by FinnGen, and a European ancestry GWAS run in UKBB using SAIGE (11,186 cases, 273,812 controls) (S3 Table). As a sensitivity analysis, all MR tests were performed using disease association

estimates based on FinnGen only, where available, to alleviate bias due to sample overlap between the exposure and outcome GWAS sources.

Power to detect MR associations was calculated using 2 methods, one designed for general 2-sample MR [48] and the other for MR performed on binary outcomes [49]. Briefly, these methods calculate power by accounting for GWAS sample size, the proportion of cases in case–control GWASs, and the variance explained by genetic instruments for the exposure. The power to detect a true odds ratio (OR) association of 1.1 or more extreme at an unadjusted significance level of 0.05 was estimated.

Instrument SNPs were extracted from the outcome GWAS results and harmonised for consistency in the alleles, and MR was performed using the TwoSample MR v0.5.4 R package [50]. Two methods for MR—IVW and MR-Egger—were evaluated, and the best method was selected via Ruecker's framework [51]. Briefly, this framework advises to choose the MR method with least heterogeneity as assessed by Cochran's $Q$ statistic, while accounting for the trade-off between power and pleiotropy [52]. IVW results, which were chosen by Ruecker's framework for all tested associations, are reported in the main text. However, as these are still susceptible to pleiotropy, results from all methods, including weighted median MR [53], are calculated for robustness and displayed in the supplementary information. Correction for multiple hypothesis testing for 24 tests in each analysis (3 exposures × 8 outcomes) was applied with the FDR method, and significance established at FDR < 0.05. MR-Egger intercept tests were performed to detect horizontal pleiotropy, and single-SNP and leave-one-out analyses were used to identify outlier SNPs driving relationships [50].

Reverse MR for obesity traits regressed on female reproductive conditions was performed as detailed above. Genetic instruments for endometriosis (14,926 cases, 189,715 controls) [54], PCOS (10,174 cases, 103,164 controls) [47], and UF (20,406 cases, 223,918 controls) [3] were constructed from index variants identified by the largest European ancestry GWAS for each trait. Instrument strength was assessed by $F$-statistics (endometriosis, 16 SNPs, $F$ = 5.13; PCOS, 14 SNPs, $F$ = 41.6; UF, 29 SNPs, $F$ = 11.1). Associations of genetic instruments for these reproductive conditions with BMI, WHR, and WHRadjBMI were obtained from female-specific summary statistics from the above-mentioned GIANT–UKBB meta-analysis [34].

## Non-linear MR

For non-linear MR analyses, we selected female UKBB participants of white British ancestry with no second-degree or closer relatives in the study, as identified by the UKBB team [55], to avoid violation of the MR assumption of random assignment of genetic variants; 207,705 women were retained following this selection. Genetic instruments for BMI were constructed for each individual using female-specific index variants from Pulit et al.'s GIANT–UKBB meta-analysis [34]. The instruments for BMI explained 4.15% of trait variance after adjustment for age, age squared, smoking status, assessment centre, genotyping array, and the first 10 genetic principal components to account for population stratification. Binomial non-linear MR, a method designed to assess genetically predicted causal relationships in different exposure strata while avoiding collider bias, was performed using the fractional polynomial method with 100 quantiles and the piecewise linear method with 10 quantiles [56]. Outcomes were restricted to female reproductive disorders with prevalence > 5% in UKBB (i.e., HMB, miscarriage, and UF) to maintain sufficient sample sizes in each quantile to estimate localised average causal estimates. We assessed non-linearity with the fractional polynomial non-linearity and Cochran's $Q$ tests, and tested heterogeneity of the instrumental variable with the Cochran's $Q$ and trend tests. All analyses were performed with the nlmr v2.0 R package [56].

## MR with mediation analysis

To investigate the extent to which obesity affects female reproductive disorders via hormone-related mediators, 2-step MR by the product of coefficients method was performed using GWAS summary statistics. This method was chosen as female reproductive disease phenotypes are binary outcomes with disease prevalence < 10% in UKBB, for which 2-step MR provides the least biased estimates of mediation [57]. Summary statistics for leptin ($N$ = 33,987) [58], fasting insulin ($N$ = 51,750) [59], and insulin sensitivity ($N$ = 16,753) [60] were obtained from publicly available European ancestry GWAS sources that do not include samples from UKBB to minimise bias from sample overlap (S1 Table).

In the first step of 2-step MR, the mediators were regressed on obesity-related exposures using summary statistics MR methods described above. The direction of causality for all relationships was confirmed with the MR-Steiger directionality test [61], and reciprocal MR with mediator instruments and obesity-related exposures as outcomes were performed to ensure correct direction of causality. In the second step, multivariable MR was performed using combined genetic instruments for each obesity trait and hormone to estimate the independent effect of the mediator on each outcome after adjusting for the value of the exposure, and to estimate the independent effect of the exposure on outcome when adjusted for the value of each mediator. This was only done for traits where the total unadjusted effect of the exposure on the outcome was significant (FDR < 0.05). ORs for binary outcomes were converted to log ORs to calculate mediated effect by the product of coefficients method. The proportion of effect mediated was calculated by dividing the indirect effect by the total effect. Standard errors were estimated with the delta method [62].

## Disease and SNP clustering

To assess similarities in the aetiological relationships of different reproductive conditions with obesity traits, we projected single SNP genetic effect estimates for BMI, WHR, and WHRadjBMI on the reproductive traits, estimated using the Wald ratio, in a 2-dimensional space using the Uniform Manifold Approximation and Projection (UMAP). Briefly, each 8-by-$M$ matrix of SNP effect estimates (for 8 female reproductive outcomes, where $M$ is 281 for BMI, 203 for WHR, and 266 for WHRadjBMI) was reduced to a 2-dimensional representation while maintaining as close a topological relationship between the 8 reproductive outcomes as possible, as measured by cross entropy [63]. SNPs were annotated to their nearest gene with SNPsnap [64].

To identify the genetic instruments driving the obesity–reproductive trait association, and identify clusters of SNPs with distinct associations, we clustered SNPs by the magnitude of their causal estimates using mixture model clustering in the MR-Clust v0.1.0 R package [65]. For each obesity trait–reproductive disease pair, the algorithm distinguishes the genetic instruments for the obesity traits that do not have an effect on the disease ('null clusters') from those that have a similar scaled effect on the disease ('substantial clusters') and those that have a scaled effect that cannot be grouped with other variants ('junk clusters').

## Research ethics

This research has been conducted using the UKBB resource under application number 11867. All procedures and data collection in UKBB were approved by the UKBB Research Ethics Committee (reference number 11/NW/0274), with participants providing full written informed consent for participation in UKBB and subsequent use of their data for approved applications. All other publicly available de-identified summary data used in this study have ethical permissions from their respective institutional review boards.

### Code availability

All scripts used in analyses are deposited at https://github.com/lindgrengroup/obesity_femrepr_MR.

## Results

### Obesity traits are observationally associated with female reproductive diseases in UKBB

BMI at baseline assessment (age 40–69 years) was positively associated with the prevalence of most female reproductive disorders in UKBB, with the strongest association observed between BMI and PCOS (OR [95% CI] per 1 SD higher BMI = 1.87 [1.80–1.94], $P = 1.90 \times 10^{-64}$). Associations with WHRadjBMI were null or lower than those for WHR (ORs for WHRadjBMI versus WHR: PCOS, 1.06 versus 1.48; pre-eclampsia, 1.02 versus 1.13; endometriosis, 1.02 versus 1.08; HMB, 1.06 versus 1.14; UF, 1.02 versus 1.08), indicating that BMI may be driving many of the associations between WHR and female reproductive diseases (Figs 1 and 2; Table 2). Infertility was the only disorder for which BMI (OR = 0.894, $P = 2.16 \times 10^{-07}$) and WHR (OR [95% CI] = 0.927 [0.884–0.969], $P = 4.08 \times 10^{-04}$) were inversely associated with disease. All associations were estimated in 257,193 women of European ancestry in UKBB, with cases defined as in Table 1 and all non-cases as controls.

Non-linear models explained the associations of BMI with many reproductive disorders better than linear models. We observed inverted-U and plateau relationships with endometriosis (linear AIC = 67,091, GAM AIC = 67,051), UF (linear AIC = 134,160, GAM AIC = 134,094), HMB (linear AIC = 116,687, GAM AIC = 116,636), miscarriage (linear AIC = 314,828, GAM AIC = 314,819), and pre-eclampsia (linear AIC = 24,826, GAM AIC = 24,814) (Fig 1; S4 Table). All 3 obesity traits displayed U-shaped relationships with PCOS.

We additionally found that having comparatively larger body size than average at age 10 years, as self-reported, was associated with increased prevalence of endometriosis (OR [95% CI] = 1.12 [1.06–1.19], $P = 2.67 \times 10^{-04}$), heavy menstrual bleeding (OR [95% CI] = 1.18 [1.14–1.22], $P = 3.43 \times 10^{-15}$), PCOS (OR [95% CI] = 1.92 [1.75–2.10], $P = 1.59 \times 10^{-13}$), and UF (OR [95% CI] = 1.07 [1.03–1.11], $P = 1.03 \times 10^{-03}$) (S6 Table). However, being thinner than average in early life was also associated with increased prevalence of endometriosis (OR [95% CI] = 1.21 [1.16–1.26], $P = 1.20 \times 10^{-13}$), heavy menstrual bleeding (OR [95% CI] = 1.10 [1.07–1.14], $P = 7.03 \times 10^{-08}$), and miscarriage (OR [95% CI] = 1.09 [1.07–1.11], $P = 4.03 \times 10^{-18}$). Observational estimates for the relationship between all obesity traits and female reproductive disorders did not differ with or without adjustment for smoking status (S5 Table). Statistical significance after multiple testing correction for 21 tests (3 exposures × 7 outcomes) was established at FDR < 0.05, unadjusted $P < 0.04$.

### Body fat distribution is genetically causally related to risk of female reproductive diseases

Two-sample MR indicated that higher genetically predicted WHR and WHRadjBMI are associated with higher risk of pre-eclampsia (OR [95% CI] per 1-SD increase in trait: WHR, 1.57 [1.16–2.10], $P = 2.92 \times 10^{-03}$; WHRadjBMI, 1.43 [1.13–1.80], $P = 2.46 \times 10^{-03}$), HMB (WHR, 1.28 [1.13–1.46], $P = 1.42 \times 10^{-04}$; WHRadjBMI, 1.16 [1.04–1.29], $P = 5.20 \times 10^{-03}$), endometriosis (WHR, 1.24 [1.05–1.47], $P = 1.00 \times 10^{-02}$; WHRadjBMI, 1.24 [1.08–1.41], $P = 1.67 \times 10^{-03}$), UF (WHR, 1.24 [1.10–1.41], $P = 6.20 \times 10^{-04}$; WHRadjBMI, 1.17 [1.06–1.29], $P = 1.95 \times 10^{-03}$), infertility (WHRadjBMI, 1.21 [1.03–1.43], $P = 2.14 \times 10^{-02}$), and PCOS

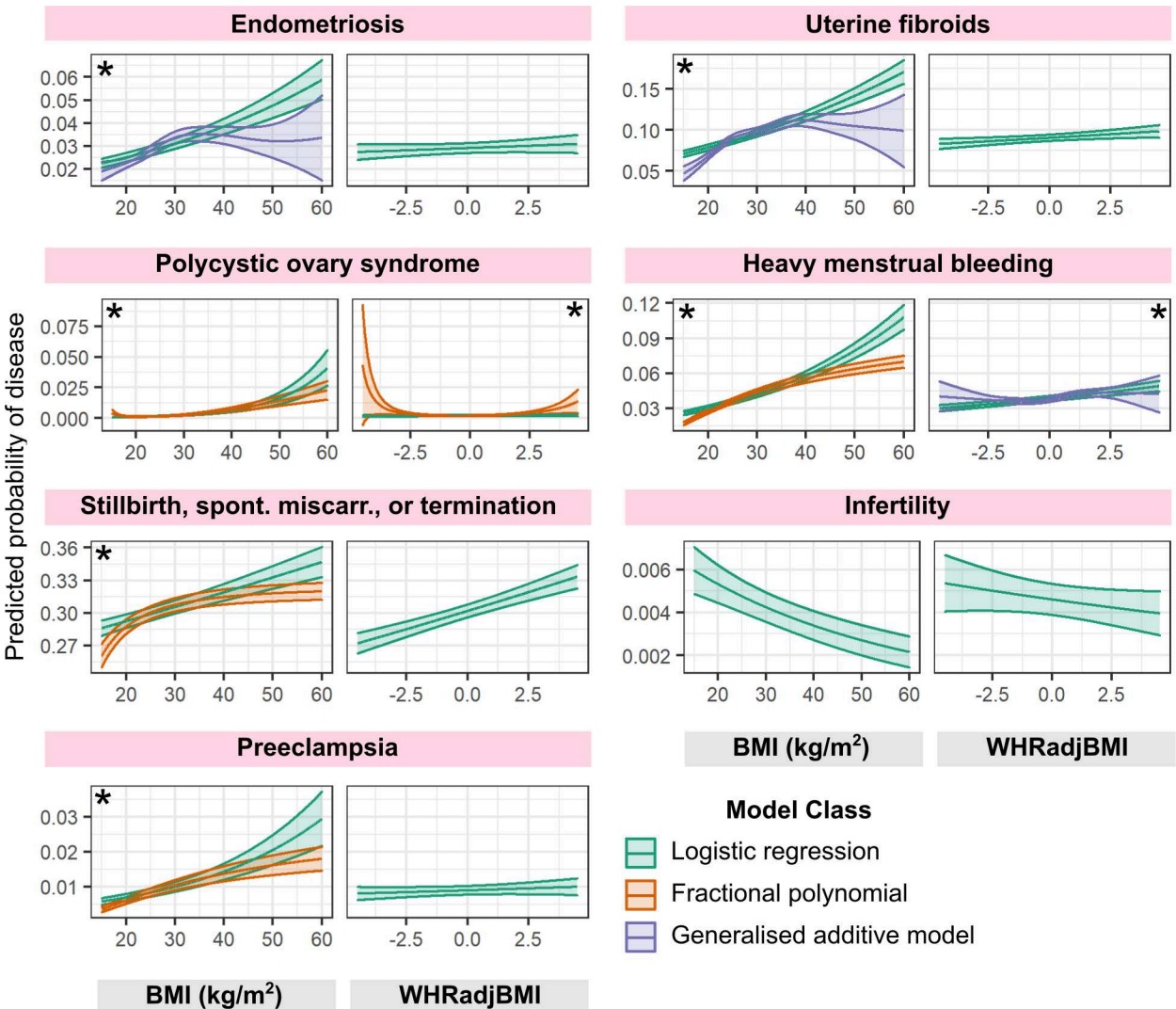

**Fig 1. Predicted probability of developing female reproductive disorders as a function of obesity-related traits.** A series of logistic regression, fractional polynomial, and generalised additive models were fitted to estimate the probability of developing female reproductive disorders as a function of obesity-related traits in UK Biobank. Predicted fits for logistic and best-fitting non-linear models were better than those for logistic regression (as evaluated with Akaike information criterion), and 95% confidence intervals about the mean are displayed. Asterisks indicate that non-linear models fit the data better than linear models. All models were adjusted for age, age squared, assessment centre, and smoking status. BMI, body mass index; spont. miscarr., spontaneous miscarriage; WHRadjBMI, waist-to-hip ratio adjusted for BMI.

(WHR, 1.07 [1.02–1.11], $P = 4.30 \times 10^{-03}$) (Table 2; Fig 2). The genetic estimates for the effect of WHR and WHRadjBMI on several reproductive disorders were higher than their observational counterparts (heterogeneity *P-het* < 0.0151 for 6/14 associations). While genetically predicted BMI was also associated with increased risk of most female reproductive disorders (ORs per 1 SD higher BMI ranged from 1.06 for sporadic miscarriage to 2.09 for pre-eclampsia), MR estimates of associations between BMI and PCOS (OR [95% CI] = 1.13 [1.08–1.19], $P = 7.60 \times 10^{-08}$) were much attenuated compared to observational results (*P-het* = $2.82 \times 10^{-30}$).

Genetically predicted VAT mass was associated with the development of pre-eclampsia (OR [95% CI] per 1-kg increase in predicted VAT mass = 3.08 [1.98–4.79], $P = 6.65 \times 10^{-07}$), PCOS (OR [95% CI] = 1.15 [1.08–1.23], $P = 3.24 \times 10^{-05}$), and HMB (OR [95% CI] = 1.32

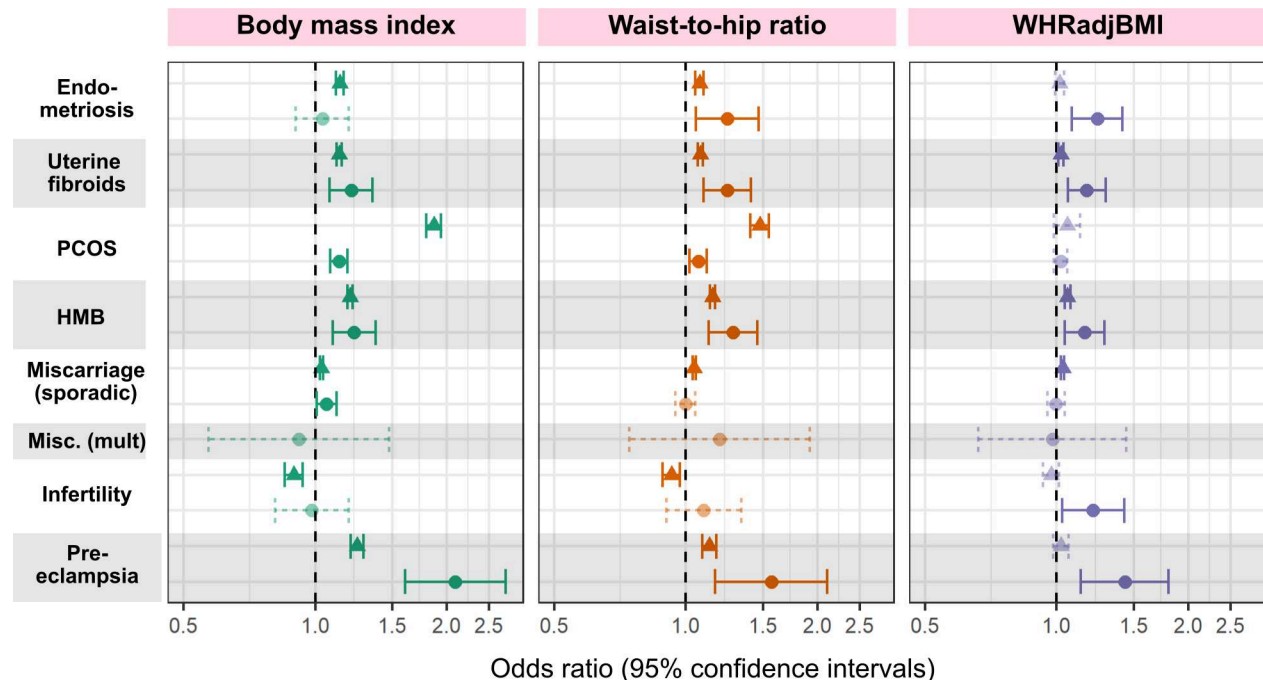

**Fig 2. Comparison of observational and genetically causal relationships between obesity-related traits and female reproductive disorders.**
Odds ratios and 95% confidence intervals per 1 SD higher obesity trait displayed. Significant relationships (false discovery rate [FDR]–adjusted *P* value < 0.05) are in solid lines while non-significant (n.s.) ones are shown with dotted lines. For observational results, BMI, WHR, and WHRadjBMI adjusted for age, age squared, region (assessment centre), and smoking status are used as predictors in a logistic regression model. Causal relationships between genetically predicted obesity-related traits and female reproductive disorders are assessed by 2-sample Mendelian randomisation (MR). The displayed method (inverse-variance-weighted) was determined via Rucker's model selection framework to minimise heterogeneity of the estimate. "Miscarriage (sporadic)" is self-reported stillbirth, spontaneous miscarriage, or termination for observational results and sporadic miscarriage for MR results. BMI, body mass index; HMB, heavy menstrual bleeding; Misc. (mult.), multiple consecutive miscarriage; PCOS, polycystic ovary syndrome; WHR, waist-to-hip ratio; WHRadjBMI, waist-to-hip ratio adjusted for BMI.

[1.06–1.64], *P* = 0.0125) (Fig 3; S7 Table). The differential association of genetically predicted body fat distribution with female reproductive traits was further reflected in the heterogeneous associations of WC and HC with disease development. Increased WC posed a higher risk than did increased HC for pre-eclampsia (OR per 1-SD increase: 1.93 for WC versus 1.40 for HC, heterogeneity *P-het* = 0.0373), HMB (1.41 for WC versus 1.12 for HC, *P-het* = $3.60 \times 10^{-03}$), UF (1.32 for WC versus 1.12 for HC, *P-het* = $7.70 \times 10^{-03}$), and PCOS (1.16 for WC versus 1.10 for HC, *P-het* = 0.0325). We did not see this heterogeneity in observational associations (all *P-het* > 0.164) (Fig 3; S12 Table).

No significant associations were found when restricting MR analyses to genetic instruments with a specific effect on waist but not HC, or on hip but not WC (S13 Table; S1 Fig), but the power of these instruments to detect ORs more extreme than 1.1 was limited to 5%–20% (S14 Table). No non-linear MR models explained the genetic associations of BMI with any reproductive disorder better than linear MR models (S4 Fig). However, the

**Table 2. Observational and genetic associations between obesity traits and female reproductive disorders.**

| Diagnosis | Obesity trait | Logistic regression | | Mendelian randomisation | | |
|---|---|---|---|---|---|---|
| | | OR (95% CI) per 1 SD higher obesity trait | P value | Number of SNPs | OR (95% CI) per 1 SD higher obesity trait | P value |
| **Endometriosis** | BMI | 1.14 (1.12–1.16) | $4.45 \times 10^{-29}$ | 264 | 1.04 (0.902–1.19) | 0.606 |
| | WHR | 1.07 (1.05–1.10) | $1.00 \times 10^{-09}$ | 190 | 1.24 (1.05–1.47) | $1.00 \times 10^{-02}$ |
| | WHRadjBMI | 1.02 (0.99–1.04) | 0.188 | 250 | 1.24 (1.08–1.41) | $1.67 \times 10^{-03}$ |
| **Heavy menstrual bleeding** | BMI | 1.20 (1.18–1.22) | $7.78 \times 10^{-117}$ | 268 | 1.23 (1.10–1.37) | $3.62 \times 10^{-04}$ |
| | WHR | 1.15 (1.13–1.16) | $8.35 \times 10^{-69}$ | 191 | 1.28 (1.13–1.46) | $1.42 \times 10^{-04}$ |
| | WHRadjBMI | 1.06 (1.04–1.07) | $5.30 \times 10^{-13}$ | 251 | 1.16 (1.04–1.29) | $5.20 \times 10^{-03}$ |
| **Infertility** | BMI | 0.894 (0.852–0.936) | $2.16 \times 10^{-07}$ | 267 | 0.982 (0.810–1.19) | 0.856 |
| | WHR | 0.927 (0.884–0.969) | $4.08 \times 10^{-04}$ | 191 | 1.10 (0.901–1.34) | 0.355 |
| | WHRadjBMI | 0.971 (0.929–1.01) | 0.176 | 251 | 1.21 (1.03–1.43) | 0.0214 |
| **Miscarriage (sporadic)** | BMI | 1.03 (1.02–1.04) | $4.28 \times 10^{-14}$ | 265 | 1.06 (1.01–1.12) | 0.0238 |
| | WHR | 1.04 (1.04–1.05) | $3.82 \times 10^{-24}$ | 190 | 0.998 (0.947–1.05) | 0.933 |
| | WHRadjBMI | 1.03 (1.02–1.04) | $1.87 \times 10^{-13}$ | 250 | 0.996 (0.953–1.04) | 0.878 |
| **Miscarriage (multiple consecutive)** | BMI | | | 254 | 0.917 (0.570–1.48) | 0.720 |
| | WHR | | | 184 | 1.20 (0.743–1.92) | 0.462 |
| | WHRadjBMI | | | 240 | 0.978 (0.662–1.44) | 0.911 |
| **Polycystic ovary syndrome** | BMI | 1.87 (1.80–1.94) | $1.90 \times 10^{-64}$ | 268 | 1.13 (1.08–1.19) | $7.60 \times 10^{-08}$ |
| | WHR | 1.48 (1.41–1.55) | $3.32 \times 10^{-26}$ | 191 | 1.07 (1.02–1.11) | $4.30 \times 10^{-03}$ |
| | WHRadjBMI | 1.06 (0.986–1.13) | 0.124 | 251 | 1.02 (0.990–1.06) | 0.222 |
| **Pre-eclampsia** | BMI | 1.25 (1.21–1.29) | $3.85 \times 10^{-25}$ | 266 | 2.09 (1.60–2.73) | $5.16 \times 10^{-08}$ |
| | WHR | 1.13 (1.09–1.17) | $4.97 \times 10^{-09}$ | 191 | 1.57 (1.16–2.10) | $2.92 \times 10^{-03}$ |
| | WHRadjBMI | 1.02 (0.982–1.07) | 0.272 | 250 | 1.43 (1.13–1.80) | $2.46 \times 10^{-03}$ |
| **Uterine fibroids** | BMI | 1.14 (1.12–1.15) | $2.43 \times 10^{-63}$ | 268 | 1.21 (1.08–1.35) | $9.93 \times 10^{-04}$ |
| | WHR | 1.08 (1.06–1.09) | $2.75 \times 10^{-23}$ | 191 | 1.24 (1.10–1.41) | $6.20 \times 10^{-04}$ |
| | WHRadjBMI | 1.02 (1.01–1.04) | $2.94 \times 10^{-03}$ | 251 | 1.17 (1.06–1.29) | $1.95 \times 10^{-03}$ |

Reported P values from logistic regression and 2-sample Mendelian randomisation, testing against the null hypothesis that association ORs are equal to 1. Logistic regression models were adjusted for age, age squared, assessment centre, and smoking status. No values are reported for logistic regression of multiple miscarriage on obesity traits as these data are not available in UK Biobank. Sporadic miscarriage results for logistic regression represent stillbirth, miscarriage, and spontaneous termination in UK Biobank.

BMI, body mass index; CI, confidence interval; OR, odds ratio; SD, standard deviation; SNP, single nucleotide polymorphism; WHR, waist-to-hip ratio; WHRadjBMI, waist-to-hip ratio adjusted for BMI.

power to detect non-linear effects was severely limited by the lower number of cases in each quantile of the BMI distribution in which analyses were run. Statistical significance after multiple testing correction for 24 tests (3 exposures × 8 outcomes) was established at FDR < 0.05, unadjusted P < 0.03.

SNPs identified in female-only GWASs and with female-specific weights for BMI, WHR, and WHRadjBMI [34] were found to be the strongest instruments, with F-statistics > 60; instrument strength for WC and HC was >45 (S1 Table). We prioritised MR results based on the IVW method over less powered MR-Egger and weighted median methods, as there was no evidence for directional pleiotropy (MR-Egger horizontal pleiotropy P > 0.0547) and the ratio of Cochran's $Q'$ (Egger) to Q (IVW) was >0.876 (S7 Table). For the 24/64 analyses for which IVW indicated a significant effect, the effect estimate of the other methods was either directionally consistent (13/24) or non-significant (11/24), but never opposite (S1 Fig). Estimates were also consistent when based only on FinnGen summary statistics (heterogeneity

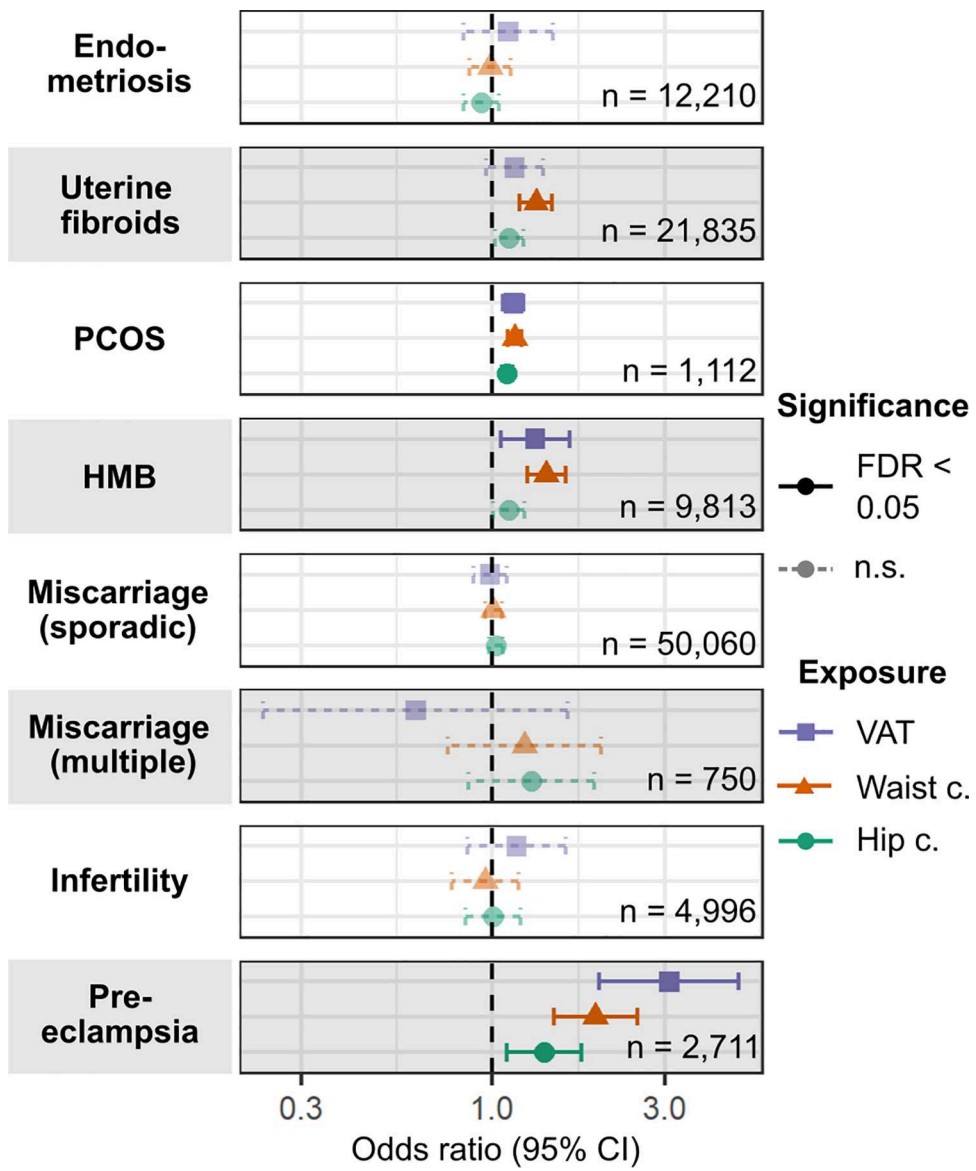

**Fig 3. Causal associations of genetically predicted visceral adipose tissue mass, waist circumference, and hip circumference with female reproductive disorders.** Odds ratios (ORs) per 1-kg increase in predicted visceral adipose tissue (VAT) mass or per 1-SD increase in waist circumference or hip circumference and 95% confidence intervals (CI) are estimated by 2-sample Mendelian randomisation. The displayed method (inverse-variance-weighted) was determined via Rucker's model selection framework. The number of cases for each disease is indicated. Significant relationships (false discovery rate [FDR]–adjusted P value < 0.05) are in solid lines while non-significant (n.s.) ones are shown with dotted lines. c., circumference; HMB, heavy menstrual bleeding; PCOS, polycystic ovary syndrome.

$P > 0.163$), with instruments from GIANT only (heterogeneity $P > 0.0541$), or with combined-sex instruments (heterogeneity $P > 0.999$), suggesting that the findings were not substantially biased due to sample overlap between exposure and outcome GWAS sources or ascertainment bias in UKBB [35,36] (S9 and S10 Tables; S1 and S2 Figs). Finally, results for WHRadjBMI did not appear to be affected by collider bias, as estimates did not differ when using WHRadjBMI GWAS instruments compared to a multivariable analysis for WHR SNPs and BMI SNPs in the same model (S11 Table; S3 Fig).

We did not find evidence for reverse causal associations of endometriosis, PCOS, or UF with BMI, WHR, and WHRadjBMI (S15 Table). However, these estimates may be biased by weak genetic instruments for endometriosis (*F*-statistic = 5.13) and UF (*F*-statistic = 11.1) and high heterogeneity for all associations (Cochran's *Q P* < $4.71 \times 10^{-06}$). We were limited in assessing the reverse causality of other female reproductive conditions on obesity traits by the lack of index SNPs in large-scale publicly available GWAS summary statistics.

## Leptin and insulin mediate the genetically predicted causal associations of obesity with female reproductive disorders

We applied a series of MR-based mediation analyses [66,67] to study the role of hormonal factors—leptin and insulin resistance—in mediating the relationships between obesity and female reproductive health (Fig 4A). The effects of BMI, WHR, and WHRadjBMI on endometriosis, PCOS, pre-eclampsia, and UF were attenuated (95% CIs of ORs all contain 1) when adjusted for leptin, fasting insulin, or insulin sensitivity as measured by the modified Stumvoll Insulin Sensitivity Index (ISI) (Fig 4B, S16 Table). Furthermore, leptin and insulin influence risk of pre-eclampsia independently of obesity. After adjustment for BMI, leptin (β ± standard error [SE] = 0.887 ± 0.232, *P* = $1.28 \times 10^{-04}$), fasting insulin (β ± SE = 1.42 ± 0.441, *P* = $1.29 \times 10^{-03}$), and ISI (β ± SE = −0.503 ± 0.169, *P* = $2.99 \times 10^{-03}$) were all associated with risk of pre-eclampsia (Fig 4C; Table 3). Similarly, fasting insulin (β ± SE = 1.27 ± 0.473, *P* = $7.15 \times 10^{-03}$) and ISI (β ± SE = −0.793 ± 0.183, *P* = $1.46 \times 10^{-05}$) had genetic associations with pre-eclampsia upon adjustment for WHR. Leptin, fasting insulin, and ISI did not have significant genetic associations with endometriosis, PCOS, or UF after adjustment for obesity traits, nor did they have any associations with the obesity traits themselves (S19 Table). Statistical significance after multiple testing correction for 48 tests (3 exposures × 4 outcomes × 4 adjustments) was established at FDR < 0.05, unadjusted *P* < 0.01.

We calculated the proportion of total obesity effect mediated by leptin, fasting insulin, and ISI for disorders where the effects of obesity traits and mediators were significant at unadjusted *P* < 0.05. We found that leptin (50.2% of the effect of BMI on pre-eclampsia), fasting insulin (27.7%–36.6%), and ISI (19.1%–50.1%) each mediated the total genetically predicted effects of obesity traits on female reproductive disorders (Table 4).

## Other metabolic and hormone pathways may drive the aetiological relationships of obesity with female reproductive diseases

We assessed the similarities in the aetiological relationships of different reproductive conditions with obesity, by projecting the single SNP genetic effect estimates for BMI, WHR, and WHRadjBMI on the reproductive traits in a 2-dimensional space using UMAP (Fig 5A). The UMAP projections based on all obesity traits clustered female reproductive diseases into three groups; one, consisting of endometriosis, UF, infertility, and HMB, which was separated from the second (sporadic and multiple consecutive miscarriage). Group 3 consisted of PCOS and pre-eclampsia, which clustered closely in UMAP plots of the effect of WHR and WHRadjBMI variants, but were separated by BMI-associated variants. This reflects a shared genetic component of the aetiological role of general and central obesity in the 3 groups of reproductive conditions.

We further examined whether different aspects of obesity play an aetiological role in different reproductive conditions. For each obesity trait–reproductive disease pair, we grouped the genetic instruments for the obesity traits by those that do not have associations with the disease ('null clusters'), those that have a similar scaled association with the disease ('substantial clusters'), and those that have a scaled association that cannot be grouped with other variants

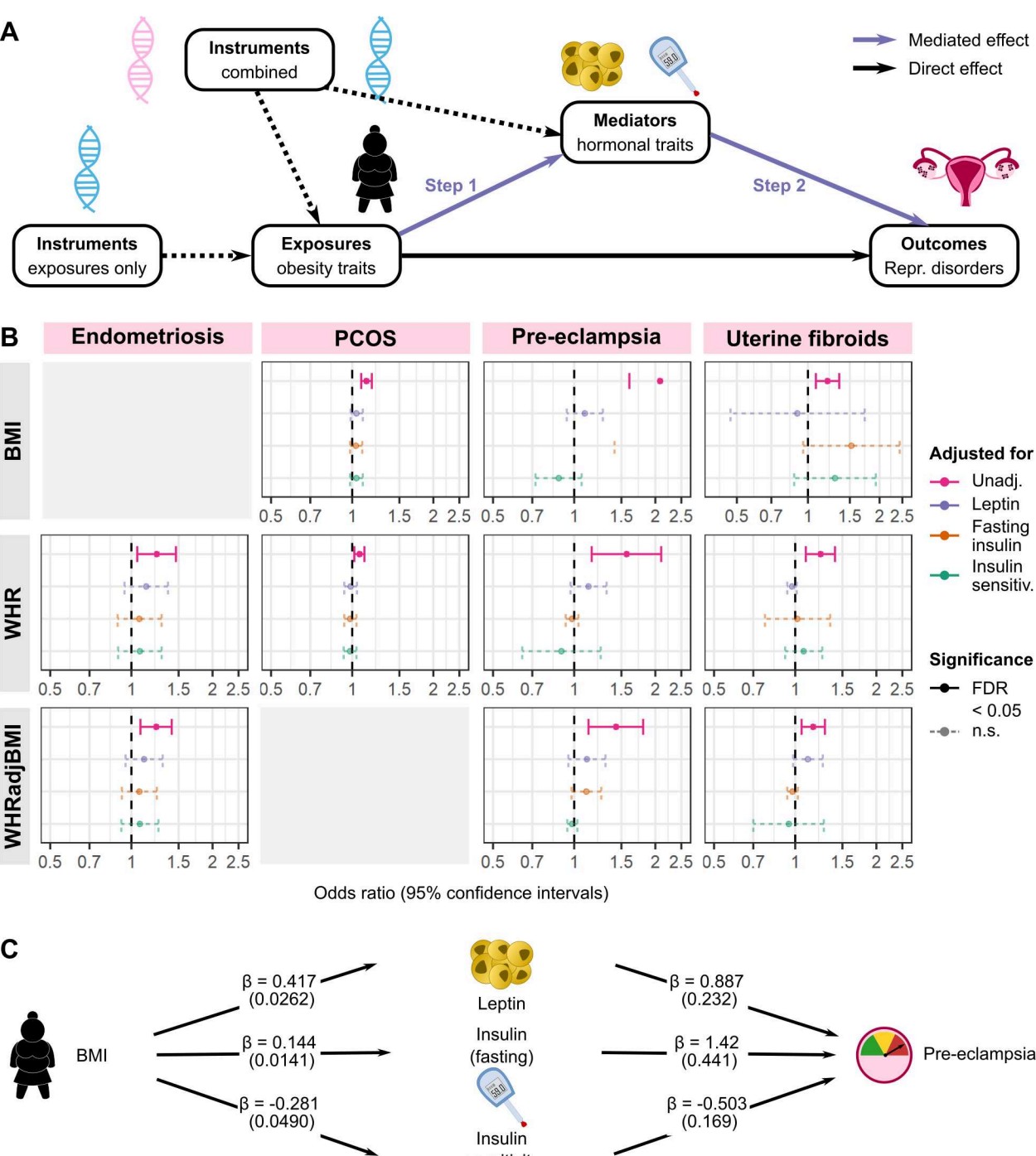

**Fig 4. Hormone mediation of genetically predicted causal effect of obesity on female reproductive disorders.** (A) Method outline for 2-step Mendelian randomisation (MR) to estimate hormonally mediated effects between obesity and female reproductive disorders. In step 1, the effect of exposures on mediators is estimated using instruments for the exposure alone, while in step 2 the independent effect of the mediators on outcomes is estimated using multivariable MR (MVMR) adjusted for exposures. All SNP–phenotype effect estimates come from different genome-wide association studies. (B) Estimated effects of obesity traits on female reproductive disorders adjusted for mediators. MVMR was performed with combined genetic instruments for each exposure–mediator combination, displayed here for relationships where unadjusted (unadj.) exposure–outcome effect was significant (false discovery rate [FDR] < 0.05). (C) Example of a mediated relationship, shown here for BMI effect on pre-eclampsia. Estimated effects for exposure–mediator (betas and standard errors) and for mediator–outcome (log odds ratios and standard errors) effects are shown. Repr., reproductive; BMI, body mass index; n.s., not significant; PCOS, polycystic ovary syndrome; sensitiv., sensitivity; WHR, waist-to-hip ratio; WHRadjBMI, waist-to-hip ratio adjusted for BMI.

**Table 3. Mendelian randomisation estimates of female reproductive disorders regressed on metabolic hormones, adjusted for obesity traits.**

| Exposure | Outcome | Obesity trait adjusted for | β ± SE per 1 SD higher exposure | P value |
|---|---|---|---|---|
| Leptin | Endometriosis | Unadjusted | −0.0745 ± 0.188 | 0.692 |
| | | BMI | −0.0457 ± 0.102 | 0.654 |
| | | WHR | −0.131 ± 0.126 | 0.297 |
| | | WHRadjBMI | −0.0964 ± 0.128 | 0.453 |
| | Uterine fibroids | Unadjusted | −0.132 ± 0.323 | 0.683 |
| | | BMI | 0.0881 ± 0.102 | 0.389 |
| | | WHR | $-4.12 \times 10^{-03} \pm 0.138$ | 0.976 |
| | | WHRadjBMI | −0.0834 ± 0.14 | 0.550 |
| | PCOS | Unadjusted | −0.134 ± 0.0701 | 0.0567 |
| | | BMI | 0.0491 ± 0.0409 | 0.231 |
| | | WHR | $8.44 \times 10^{-03} \pm 0.0476$ | 0.859 |
| | | WHRadjBMI | −0.0310 ± 0.0473 | 0.511 |
| | Pre-eclampsia | Unadjusted | 0.181 ± 0.501 | 0.718 |
| | | BMI | 0.887 ± 0.232 | $1.28 \times 10^{-04}$ |
| | | WHR | 0.468 ± 0.309 | 0.129 |
| | | WHRadjBMI | −0.102 ± 0.333 | 0.760 |
| Fasting insulin | Endometriosis | Unadjusted | 0.150 ± 0.247 | 0.544 |
| | | BMI | 0.0996 ± 0.209 | 0.634 |
| | | WHR | 0.186 ± 0.254 | 0.465 |
| | | WHRadjBMI | 0.423 ± 0.239 | 0.0770 |
| | Uterine fibroids | Unadjusted | 0.0263 ± 0.342 | 0.939 |
| | | BMI | 0.311 ± 0.183 | 0.090 |
| | | WHR | 0.313 ± 0.209 | 0.135 |
| | | WHRadjBMI | 0.262 ± 0.202 | 0.195 |
| | PCOS | Unadjusted | −0.0334 ± 0.158 | 0.833 |
| | | BMI | 0.0283 ± 0.0772 | 0.714 |
| | | WHR | 0.0932 ± 0.0788 | 0.237 |
| | | WHRadjBMI | 0.0289 ± 0.0726 | 0.690 |
| | Pre-eclampsia | Unadjusted | −0.0379 ± 0.873 | 0.965 |
| | | BMI | 1.42 ± 0.441 | $1.29 \times 10^{-03}$ |
| | | WHR | 1.27 ± 0.473 | $7.15 \times 10^{-03}$ |
| | | WHRadjBMI | 0.476 ± 0.467 | 0.308 |
| Insulin sensitivity | Endometriosis | Unadjusted | −0.0940 ± 0.133 | 0.478 |
| | | BMI | −0.101 ± 0.0864 | 0.244 |
| | | WHR | −0.131 ± 0.100 | 0.192 |
| | | WHRadjBMI | −0.102 ± 0.106 | 0.336 |
| | Uterine fibroids | Unadjusted | −0.223 ± 0.137 | 0.103 |
| | | BMI | −0.0951 ± 0.0716 | 0.184 |
| | | WHR | −0.202 ± 0.0830 | 0.0151 |
| | | WHRadjBMI | −0.120 ± 0.0855 | 0.165 |
| | PCOS | Unadjusted | −0.0239 ± 0.0616 | 0.698 |
| | | BMI | −0.0257 ± 0.0290 | 0.376 |
| | | WHR | −0.0355 ± 0.0293 | 0.226 |
| | | WHRadjBMI | −0.0443 ± 0.0294 | 0.132 |
| | Pre-eclampsia | Unadjusted | −0.354 ± 0.295 | 0.229 |
| | | BMI | −0.503 ± 0.169 | $2.99 \times 10^{-03}$ |
| | | WHR | −0.793 ± 0.183 | $1.46 \times 10^{-05}$ |
| | | WHRadjBMI | −0.519 ± 0.205 | 0.0110 |

*P* values calculated from multivariable Mendelian randomisation, testing against the null hypothesis of effect estimate (beta) equal to 0. Unadjusted estimates are from 2-sample MR with SNPs for exposures only, while adjusted estimates are from multivariable MR with SNPs for exposures and the adjustment factor, i.e., BMI, WHR, or WHRadjBMI.

BMI, body mass index; PCOS, polycystic ovary syndrome; SD, standard deviation; SE, standard error; WHR, waist-to-hip ratio; WHRadjBMI, waist-to-hip ratio adjusted for BMI.

**Table 4. Proportion of effect mediated for exposure–mediator–outcome relationships.**

| Exposure | Mediator | Outcome | Log OR (SE) per 1 SD higher exposure, *P* value | | | Proportion of effect mediated (95% CI) |
|---|---|---|---|---|---|---|
| | | | Exposure–outcome | Mediator–outcome | Exposure–mediator | |
| BMI | Leptin | Pre-eclampsia | 0.737 (0.135), $P = 5.16 \times 10^{-08}$ | 0.887 (0.232), $P = 1.28 \times 10^{-04}$ | 0.417 (0.0262), $P = 7.94 \times 10^{-57}$ | 50.2% (18.2%–82.2%) |
| | Fasting insulin | | | 1.42 (0.441), $P = 1.29 \times 10^{-03}$ | 0.144 (0.0141), $P = 1.61 \times 10^{-24}$ | 27.7% (7.40%–48.0%) |
| | Insulin sensitivity | | | −0.503 (0.169), $P = 2.99 \times 10^{-03}$ | −0.281 (0.0490), $P = 9.83 \times 10^{-09}$ | 19.1% (3.33%–35.0%) |
| WHR | Fasting insulin | Pre-eclampsia | 0.449 (0.151), $P = 2.92 \times 10^{-03}$ | 1.27 (0.473), $P = 7.15 \times 10^{-03}$ | 0.129 (0.0159), $P = 5.04 \times 10^{-16}$ | 36.6% (0%–73.7%) |
| | Insulin sensitivity | | | −0.793 (0.183), $P = 1.46 \times 10^{-05}$ | −0.283 (0.0601), $P = 2.44 \times 10^{-06}$ | 50.1% (4.98%–95.3%) |
| | Insulin sensitivity | Uterine fibroids | 0.218 (0.0637), $P = 6.20 \times 10^{-04}$ | −0.202 (0.083), $P = 1.51 \times 10^{-02}$ | −0.283 (0.0601), $P = 2.44 \times 10^{-06}$ | 26.2% (0%–54.4%) |
| WHRadjBMI | Insulin sensitivity | Pre-eclampsia | 0.358 (0.118), $P = 2.46 \times 10^{-03}$ | −0.519 (0.205), $P = 1.14 \times 10^{-02}$ | −0.168 (0.0427), $P = 8.21 \times 10^{-05}$ | 24.4% (0%–51.8%) |

*P* values calculated from 2-sample (exposure–outcome, exposure–mediator) or multivariable (mediator–outcome) Mendelian randomisation, testing against the null hypothesis of effect estimate (beta) equal to 0.

BMI, body mass index; CI, confidence interval; OR, odds ratio; SD, standard deviation; SE, standard error; WHR, waist-to-hip ratio; WHRadjBMI, waist-to-hip ratio adjusted for BMI.

('junk clusters') using MRClust [65]. One substantial cluster was identified for each pair of obesity traits and reproductive conditions. The only exception to this was the pair WHRadjBMI and UF, for which 2 substantial clusters were identified, one with positive genetic association and the other with negative association (S5 Fig; S17 Table). Of the 4 SNPs in the negative effect cluster, rs2277339 (missense variant in *PRIM1* and upstream of *HSD17B6*, involved in steroid biosynthesis) is associated with primary ovarian insufficiency, early menopause, and PCOS [68,69], and rs11694173 is intronic to *THADA*, which is also associated with PCOS [47]. On the other hand, 4 of 10 SNPs in the positive effect cluster are associated with metabolic traits—rs12328675 and rs2459732 are associated with circulating leptin [70], rs6905288 is associated with type 2 diabetes and thyroid stimulating hormone [71,72], and rs4686696 is intronic to insulin-like growth factor *IGF2BP2*.

SNPs with high probability of belonging to the substantial cluster (≥80% probability) were generally unique to each obesity–disease relationship, with no more than 2 variants shared between any 2 clusters (Fig 5B). However, 6 BMI index SNPs had positive genetic estimates for both PCOS and pre-eclampsia, including rs1121980 in the adipose-associated gene *FTO* and rs7498665 in *SH2B1*, linked to insulin resistance in obesity. The BMI-associated variant rs7084454 (intronic to *MLLT10*) was shared by substantial clusters for PCOS, endometriosis, and UF, while rs114760566 (mapped to *HMGA1*, associated with type 2 diabetes and multiple lipomatosis) was shared by endometriosis and UF. We evaluated the biological effect of the top SNPs in each substantial cluster with the DEPICT algorithms for pathway enrichment and gene prioritisation. We recapitulated the known association of the GEMIN5 subnetwork with PCOS in SNPs causal for BMI–PCOS [73]. Gene prioritisation for WHR–endometriosis causal SNPs highlighted *TBX15*, an important mesodermal transcription factor with roles in endometrial and ovarian cancer [74,75].

## Discussion

In this systematic genetics-based causal investigation of the aetiological role of obesity in female reproductive health, we report evidence that common indices of obesity are associated

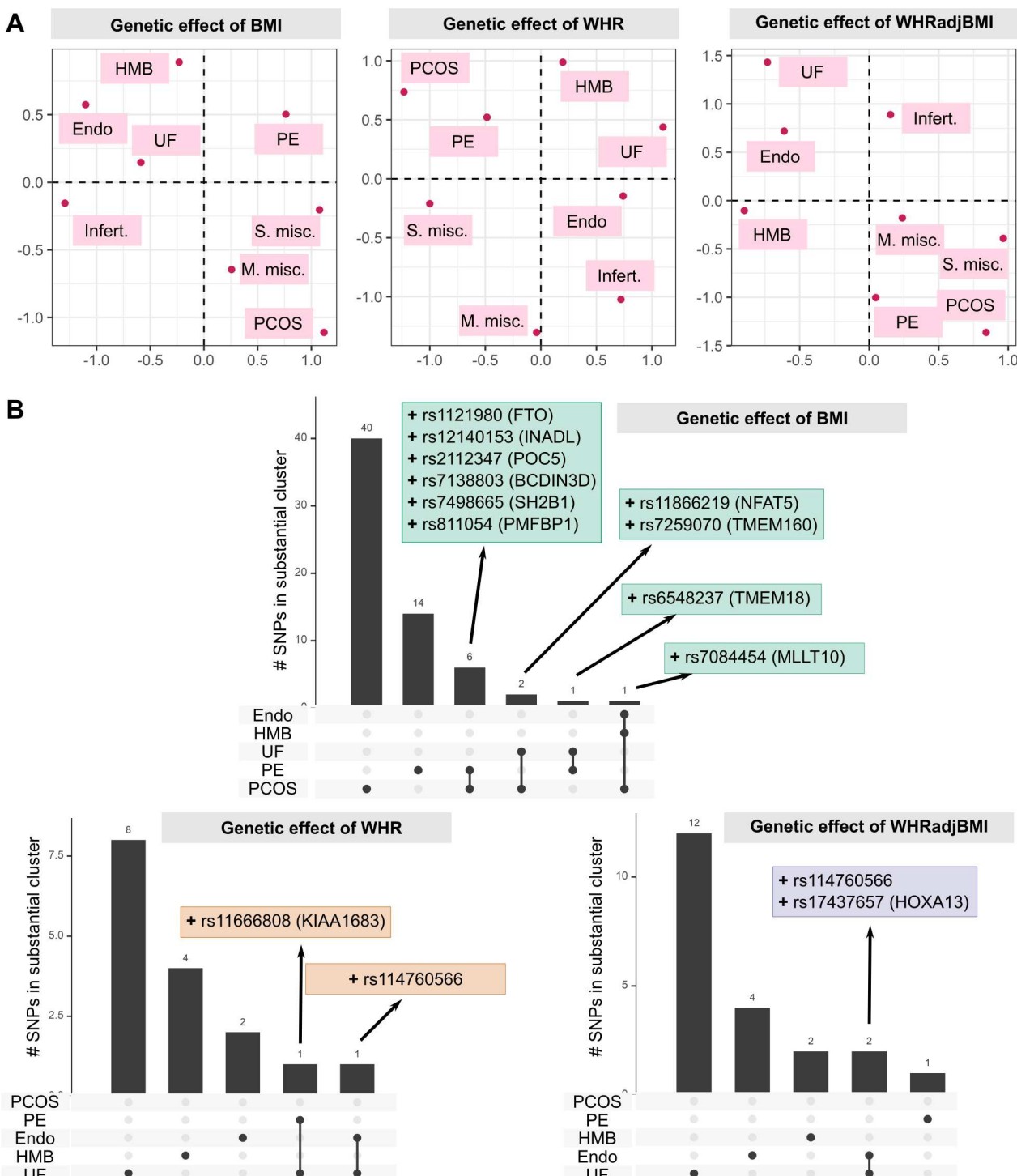

**Fig 5. Clustering of the genetic effect of obesity on female reproductive disorders.** (A) Female reproductive disorders separated by the genetic effects of different obesity traits on reproductive conditions. Uniform Manifold Approximation and Projection (UMAP) was used to plot the separation of diseases based on genetic effects of obesity instruments, evaluated by 2-sample Mendelian randomisation. (B) Number of SNPs in substantial clusters for each obesity × female reproductive condition relationship. SNPs with ≥80% probability of belonging to a substantial cluster (MRClust) are displayed with their nearest gene as annotated by SNPsnap and '+' or '−' representing causal effect direction. BMI, body mass index; Endo, endometriosis; HMB, heavy menstrual bleeding; Infert., infertility; M. misc., multiple consecutive miscarriage; PCOS, polycystic ovary syndrome; PE, pre-eclampsia; S. misc., sporadic miscarriage; UF, uterine fibroids; WHR, waist-to-hip ratio; WHRadjBMI, waist-to-hip ratio adjusted for BMI.

with increased risk of a broad range of reproductive conditions, and these associations may be non-uniform across the obesity spectrum. The strongest association of generalised obesity was found with pre-eclampsia, while more modest associations were observed for nearly all other studied conditions. We identified endocrine mechanisms, including those related to leptin and insulin resistance, as potential drivers of aetiological relationships of both generalised and central obesity with female reproductive health. Finally, we found genetic evidence that certain groups of reproductive conditions, such as UF and endometriosis, may share a mechanistically similar relationship with obesity.

The results from our MR investigations are less likely to be biased by confounding or reverse causation than observational epidemiological results. We undertook multiple supplementary and sensitivity analyses to evaluate the plausibility of instrumental variable assumptions and robustness to horizontal pleiotropy, outliers, collider bias, and sample overlap that may invalidate or bias MR estimates. While causal associations from MR must be interpreted with caution as several assumptions of the method are untestable, the concordance of our estimates from different methods and analytical approaches indicates strong support for a causal role of obesity in the aetiology of female reproductive conditions.

Our findings highlight that the relationships between obesity and female reproductive disorders are (i) non-uniform in their nature and strength and (ii) observationally non-linear across the obesity spectrum. We report substantial differences in the genetically predicted causal associations of BMI with reproductive diseases, with each 1-SD increase in BMI associated with double the risk of pre-eclampsia, but more moderately (ORs = 1.01–1.25 for PCOS, miscarriage, UF, and HMB) or not at all (infertility and endometriosis) affecting other conditions. Conversely, central fat distribution independent of BMI showed substantial genetically predicted effects on both infertility and endometriosis (ORs per 1-SD increase in WHRadjBMI = 1.21–1.46) as well as on pre-eclampsia and UF (ORs = 1.17–1.43), but not on PCOS, HMB, and miscarriage. These findings highlight that the aetiological role of obesity in female reproductive diseases is heterogeneous in its effect strength, and may be driven by overall adiposity (PCOS, HMB, and miscarriage), by isolated central obesity (infertility and endometriosis), or by both generalised and central obesity (pre-eclampsia and UF).

For several reproductive conditions, we found substantial differences between the observational and genetically predicted causal estimates, which may indicate a bidirectional relationship between obesity and reproductive health. For instance, while the observational analyses suggested an 87% increase in PCOS risk per 1 SD higher BMI, the MR analyses indicated that each 1 SD higher genetically predicted BMI was associated with an increased PCOS risk of only 13%. Similarly, 1-SD increases in genetically predicted WHR and WHRadjBMI were associated with a 24% increase in endometriosis risk, while the observational analyses suggest more modest increases in risk of 7% and 2%, respectively. This discrepancy may in part be due to reverse causality, which we were not powered to detect in this study, as the available genetic instruments for reproductive conditions are substantially fewer in number and weaker than those for BMI and WHR. The obesity traits upon which the observational analyses were based were measured at ages 40–69 years, which was for most conditions likely to be several years or decades after women developed the condition, and often post-menopause. While our observational analyses adjusted for the effect of age on obesity traits, adjusting for menopause status proved to be unreliable as up to 42% of women with reproductive diseases in UKBB—some of whom had undergone hysterectomies—were unsure of their menopause status, as opposed to 16% of female participants without a diagnosis of any of the studied conditions. The observational estimates may therefore capture both the effect of obesity on disease risk as well as any downstream effects of the disease or commonly used treatments on body weight and fat

distribution. For instance, the large observational effect of BMI on PCOS prevalence may reflect both a causal association of obesity with disease risk [76], as captured by the genetically predicted effect, as well as weight gain as a consequence of PCOS [77]. Other potential contributing factors to the differences between genetic and observational estimates are confounding by unmeasured variables, leading to inflated observational associations [78,79]; referral bias, wherein obesity status affects the likelihood of receiving a diagnosis [80,81]; and differences in pre- and post-menopausal weight and body fat distribution not captured by age [82].

While the observational relationships between obesity and some female reproductive disorders were non-linear, we did not find non-linearity in the genetically predicted effects of BMI on these diseases. The non-linear MR analyses were likely underpowered to detect associations, with few cases in each quantile of the BMI spectrum. However, as current GWAS approaches are focused on identifying genetic determinants of BMI across the full BMI spectrum, it is possible that the instruments used here do not capture genetic factors that specifically explain variations in BMI among those with lower (20–25 kg/m$^2$) or higher (40–45 kg/m$^2$) BMI. There is currently no evidence for this, but if this were the case, then our analysis would not identify non-linear causal associations.

We noted that genetic estimates for the effect of fat distribution were not similarly attenuated when compared to BMI effects. This disparity may be due to the differing impacts of overall and abdominal (central) adiposity, as the latter is thought to be biologically more directly linked to female reproductive health than generalised obesity, via pathways including insulin resistance and hyperandrogenaemia [5,15,16,83]. Supporting the stronger effect of central body fat, we also reported greater genetic effect estimates of WC than HC with HMB, PCOS, pre-eclampsia, and UF. Genetically predicted VAT mass was associated with increased risk of PCOS and pre-eclampsia, in line with observational studies [84]. VAT mass is also observationally associated with UF [16], yet we did not find a significant association of genetically predicted VAT mass with development of UF, which may suggest a bidirectional or reverse causal relationship.

Endometriosis and infertility were the only reproductive conditions that did not show a consistently positive link with obesity. The modest observational associations of both BMI and WHR with higher endometriosis prevalence in UKBB contradict previous studies, including prospective cohort studies, which reported that lower BMI was associated with increased disease prevalence [14,85,86]. The positive association with endometriosis may in part be due to weight gain as a consequence of the disease, for instance due to hormonal treatments [87–89], chronic pain [90], inflammation [91], or earlier onset of menopause [92]. We however did not find evidence that generalised obesity plays a causal role in the aetiology of endometriosis, which suggests that the observational finding reflects a reverse causal relationship. Indeed, we noted that being thinner than average at age 10 years posed a higher risk for development of endometriosis than did having comparatively larger body size, although both were associated with higher prevalence of disease compared to those who reported average body size in early life. Conversely, the positive genetically predicted effect of WHRadjBMI on endometriosis risk indicates a causal role for abdominal fat distribution. For infertility, we observed a similar divergence between the observational and genetically predicted effects of obesity traits, with BMI showing a negative observational association, but WHRadjBMI a genetically predicted positive association. The causes of female infertility are multiple, ranging from PCOS [93] and anovulation [94] to tubal disease [95], endometriosis [96], low oocyte quality [97], hormonal and immunological dysfunction [98–101], and yet unknown mechanisms. Each of these may have distinct and complex relationships with obesity, which cannot be captured by studying the links with infertility of any cause. Non-linear effects, such as the increased association of

under- and overweight with incidence of infertility [12,102], may also obscure these estimates, although our observational analyses did not provide evidence for a non-linear relationship.

We conducted one of the first genetics-based investigations of the mediating hormonal pathways underlying the causal relationships between obesity and female reproductive health. We identify mechanisms related to insulin resistance and leptin as mediators of the effects of obesity traits on UF and pre-eclampsia. The latter is consistent with hypotheses that obese women with metabolic dysregulation are at highest risk of developing hypertensive disorders of pregnancy via angiogenic and pro-inflammatory mechanisms. Increased circulating leptin may have a vasoconstrictive, hypertensive effect, which may be worsened by attenuation of insulin-induced vasorelaxation and increased levels of TNF-alpha and IL6 [7,22].

Finally, genetic clustering of female reproductive conditions revealed evidence supporting common genetic causes for the effects of obesity on endometriosis, UF, and HMB, which are known to share mechanisms of development [3,103]. The projection of infertility with these diseases merits following up on the genetic basis of endometriosis-related infertility, with an eye to prevention and treatment. The main strength of our work is the systematic approach to characterising the relationship between a broad range of obesity traits and common female reproductive conditions using both observational and genetic approaches. All observational associations were estimated in the same large-scale cohort study, which tends to lead to less biased estimates than case–control studies, upon which most previous results were based. Moreover, to the best of our knowledge, we have conducted the first genetics-based mediation analyses to pinpoint the mechanisms driving the causal association of obesity with reproductive diseases.

Reproductive conditions remain underdiagnosed and underreported in the UK, which was reflected in their low prevalence among female UKBB participants (Table 1). Although we based our case definitions on 3 distinct sources, i.e., participants' responses to interviews and structured surveys, primary care electronic health records dating back to 1938 at the earliest, and secondary care hospital in-patient records dating back to 1981 at the earliest, a participant's diagnosis may have been missed if, for example, the diagnostic code was not entered, the diagnosis was made in a setting where electronic health records were not implemented, or the participant could not recall ever having received such a diagnosis at baseline assessment. The low prevalence of female reproductive disorders posed a limitation to our analyses in UKBB by reducing power to identify significant associations. For this reason, we opted to use broad case categories, such as infertility of any cause, as we had insufficient power and information to examine conditions by subtypes. We also restricted our analyses to women of European ancestry, due to a lack of genetic data on women of other ancestries. Many of the reproductive diseases included here, with UF being the most notable example [104,105], are more prevalent in non-European populations, and our results may not be transferable to women of other ancestries [88,106,107], which highlights the urgent need to set up large-scale studies similar to UKBB with participants of non-European ancestry. We were further limited in investigations of metabolic, hormonal, and inflammatory mediating mechanisms by a lack of publicly available GWAS summary statistics for these traits. Finally, the lack of data on BMI and WHR prior to disease onset, and limited information on the age at which reproductive conditions were first diagnosed, complicated the interpretation of our findings from observational analyses in UKBB.

Key priorities for the future are the further exploration and validation of the pathways through which obesity increases the risk of female reproductive disease. Notably, our finding that insulin resistance may be an important mediating mechanism warrants further attention, as affordable and safe treatments are available to increase insulin sensitivity. This is demonstrated by the successful use of metformin treatment in women with PCOS [108], but such a

treatment strategy has not yet been explored for other reproductive conditions linked to obesity. More generally, better and more detailed diagnostic information on reproductive health in large-scale cohort studies is urgently required for future research on the causes, consequences, and aetiological mechanisms of female reproductive illnesses.

In conclusion, we provide genetic evidence that both generalised and central obesity play an aetiological role in a broad range of female reproductive conditions, but the extent of this link differs substantially between conditions. Our findings also highlight the importance of hormonal pathways, notably those involving leptin and insulin resistance, as mediating mechanisms and potential targets for intervention in the treatment and prevention of common female reproductive conditions.

## Supporting information

**S1 Checklist. STROBE-MR guideline.**
(PDF)

**S1 Fig. Comparison of different Mendelian randomisation methods for estimating the causal effect of obesity traits on female reproductive disorders.**
(TIF)

**S2 Fig. Mendelian randomisation sensitivity analyses for female reproductive disorders regressed on obesity traits.** (A) Female-specific genetic instruments versus combined-sex genetic instruments. (B) Reproductive outcomes from meta-analysis of UK Biobank and FinnGen summary statistics versus FinnGen only. (C) Genetic instruments from GWASs including UK Biobank participants (Pulit et al. 2019 [34]) versus external genetic instruments from GWASs without UK Biobank participants (GIANT releases, 2015).
(TIF)

**S3 Fig. Comparing the causal effect of WHR adjusted for BMI on female reproductive disorders using 2 methods.**
(TIF)

**S4 Fig. Non-linear genetic effects of obesity traits on female reproductive disorders.** (A) Non-linear Mendelian randomisation estimates for relationships between BMI and female reproductive disorders. (B) Heterogeneity across BMI spectrum in instrument variables used for non-linear Mendelian randomisation.
(TIF)

**S5 Fig. Single-SNP genetic effect estimates for obesity instruments on female reproductive disorders.**
(TIF)

**S1 Table. Genetic instrument source and strength for obesity-related and hormone exposures used in Mendelian randomisation analyses.**
(XLSX)

**S2 Table. Summary statistics sources for female reproductive outcomes used in Mendelian randomisation analyses.**
(XLSX)

**S3 Table. Datasets included in meta-genome-wide association study for polycystic ovary syndrome.**
(XLSX)

**S4 Table. Comparing model fits with Akaike information criterion to estimate the linear and non-linear effects of obesity traits on female reproductive conditions in UK Biobank.**
(XLSX)

**S5 Table. Comparing smoking adjustment in logistic regression of female reproductive conditions on obesity-related traits in UK Biobank.**
(XLSX)

**S6 Table. Logistic regression of female reproductive conditions on comparative body size at age 10 years (self-reported) in UK Biobank.**
(XLSX)

**S7 Table. Comparing causal effects of obesity traits on female reproductive conditions estimated by different 2-sample Mendelian randomisation methods.**
(XLSX)

**S8 Table. Two-sample Mendelian randomisation estimates of combined-sex instruments for BMI, WHR, and WHRadjBMI effects on female reproductive conditions.**
(XLSX)

**S9 Table. Two-sample Mendelian randomisation estimates of GIANT 2015 instruments for BMI, WHR, and WHRadjBMI effects on female reproductive conditions.**
(XLSX)

**S10 Table. Two-sample Mendelian randomisation estimates of BMI, WHR, and WHRadjBMI effects on female reproductive conditions from FinnGen only.**
(XLSX)

**S11 Table. Comparing causal effect of WHR adjusted for BMI on female reproductive conditions using WHRadjBMI GWAS instruments versus multivariable Mendelian randomisation for WHR and BMI in same model.**
(XLSX)

**S12 Table. Observational and genetically predicted causal associations of hip circumference and waist circumference with female reproductive conditions.**
(XLSX)

**S13 Table. Two-sample Mendelian randomisation effect estimates of waist-specific WHR and hip-specific WHR on female reproductive conditions.**
(XLSX)

**S14 Table. Power calculations for all 2-sample Mendelian randomisation analyses.**
(XLSX)

**S15 Table. Reciprocal Mendelian randomisation for effect of female reproductive conditions on BMI, WHR, and WHRadjBMI.**
(XLSX)

**S16 Table. Multivariable Mendelian randomisation effect estimates of obesity-related traits on female reproductive conditions, adjusted for metabolic hormones.**
(XLSX)

**S17 Table. Cluster classification and probabilities for each SNP in obesity instruments, with Mendelian randomisation effect estimates on female reproductive conditions.**
(XLSX)

**S18 Table. Step 1 of mediation Mendelian randomisation–2-sample Mendelian randomisation effect estimates of obesity-related exposures on metabolic hormones.**
(XLSX)

**S19 Table. Reciprocal Mendelian randomisation for step 1 of mediation–2-sample Mendelian randomisation effect estimates of metabolic hormones on obesity-related traits.**
(XLSX)

## Acknowledgments

We acknowledge the participants and investigators of FinnGen for their contribution to this study. This research was conducted using the UK Biobank Resource under application number 10844.

The views expressed are those of the authors and not necessarily those of the National Health Service, the National Institute for Health Research, or the Department of Health.

## Author Contributions

**Conceptualization:** Teresa Ferreira, Cecilia M. Lindgren, Laura B. L. Wittemans.

**Data curation:** Samvida S. Venkatesh, Teresa Ferreira, Nilufer Rahmioglu, Laura B. L. Wittemans.

**Formal analysis:** Samvida S. Venkatesh.

**Funding acquisition:** Cecilia M. Lindgren.

**Investigation:** Samvida S. Venkatesh, Christian M. Becker, Ingrid Granne, Krina T. Zondervan.

**Methodology:** Samvida S. Venkatesh, Stefania Benonisdottir, Michael V. Holmes.

**Resources:** Nilufer Rahmioglu.

**Supervision:** Teresa Ferreira, Cecilia M. Lindgren, Laura B. L. Wittemans.

**Validation:** Samvida S. Venkatesh.

**Visualization:** Samvida S. Venkatesh.

**Writing – original draft:** Samvida S. Venkatesh, Laura B. L. Wittemans.

**Writing – review & editing:** Samvida S. Venkatesh, Teresa Ferreira, Stefania Benonisdottir, Nilufer Rahmioglu, Christian M. Becker, Ingrid Granne, Krina T. Zondervan, Michael V. Holmes, Cecilia M. Lindgren, Laura B. L. Wittemans.

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
