## [Editor Report · Decision Letter 0]

5 Jun 2021

Dear Dr Venkatesh, 

Thank you for submitting your manuscript entitled "The role of obesity in female reproductive conditions: A Mendelian Randomisation Study" for consideration by PLOS Medicine.

Your manuscript has now been evaluated by the PLOS Medicine editorial staff and I am writing to let you know that we would like to send your submission out for external peer review.

Please re-submit your manuscript within two working days, i.e. by Jun 09 2021 11:59PM.

Kind regards,

Louise Gaynor-Brook, MBBS PhD

Associate Editor

PLOS Medicine

---

## [Decision Letter · Decision Letter 1]

17 Aug 2021

Dear Dr. Venkatesh,

Thank you very much for submitting your manuscript "The role of obesity in female reproductive conditions: A Mendelian Randomisation Study" (PMEDICINE-D-21-02467R1) for consideration at PLOS Medicine. 

Your paper was evaluated by three independent reviewers, including a statistical reviewer, and has been discussed with an academic editor with relevant expertise and among all the editors here. The reviews are appended at the bottom of this email and any accompanying reviewer attachments can be seen via the link below:

[LINK]

In light of these reviews, I am afraid that we will not be able to accept the manuscript for publication in the journal in its current form, but we would like to consider a revised version that addresses the reviewers' and editors' comments. Obviously we cannot make any decision about publication until we have seen the revised manuscript and your response, and we plan to seek re-review by one or more of the reviewers. 

We expect to receive your revised manuscript by Sep 07 2021 11:59PM. Please email us (plosmedicine@plos.org) if you have any questions or concerns.

We look forward to receiving your revised manuscript. 

Sincerely,

Louise Gaynor-Brook, MBBS PhD

Associate Editor 

PLOS Medicine

plosmedicine.org

General comments:

Throughout the paper, please adapt reference call-outs to the following style: "... abnormal menstrual bleeding [1,2]..." (noting the absence of spaces within the square brackets).

Throughout your manuscript, please substitute ‘causal effects’ with ‘casual associations’ or ‘causal relationships’

Competing Interests: Please confirm that all disclosed competing interests are outside the submitted work.

Title: Please revise your title according to PLOS Medicine's style. We suggest “Obesity and risk of female reproductive disorders: A Mendelian randomisation study” or similar

Abstract Background: The final sentence should clearly state the study question.

Abstract Methods and Findings:

Please provide brief demographic details of the study population (e.g. sex, age, ethnicity, etc)

Please include the number of individuals included in your study.

Line 33 - please be more specific as to which conditions causal effect estimates of WHR and WHRadjBMI, but not BMI, were attenuated, and quantify those associations 

Line 35 - please revise to causally associated with

In the last sentence of the Abstract Methods and Findings section, please describe 2-3 of the main limitation(s) of the study's methodology.

Abstract Conclusions:

Please begin your Abstract Conclusions with "In this study, we observed ..." or similar, to summarize the main findings from your study without overstating your conclusions.

Please temper assertions of primacy by adding ‘to the best of our knowledge’ or similar

Line 44 - please be more specific that metabolic hormones were only associated with pre-eclampsia, as this was the only association presented in the results. 

Author Summary:

In the final bullet point of ‘What Do These Findings Mean?’, please describe the main limitations of the study in non-technical language.

Introduction:

Line 49 - please be more specific on the associations between BMI and gynaecological conditions i.e. higher vs lower

Line 73 - please be more specific re adverse pregnancy outcomes

Line 78 - Please temper assertions of primacy by adding ‘to the best of our knowledge’ or similar

Methods:

Did your study have a prospective protocol or analysis plan? Please state this (either way) early in the Methods section. If a prospective analysis plan (from your funding proposal, IRB or other ethics committee submission, study protocol, or other planning document written before analyzing the data) was used in designing the study, please include the relevant prospectively written document with your revised manuscript as a Supporting Information file to be published alongside your study, and cite it in the Methods section. A legend for this file should be included at the end of your manuscript. If no such document exists, please make sure that the Methods section transparently describes when analyses were planned, and if/when reported analyses differed from those that were planned. Changes in the analysis-- including those made in response to peer review comments-- should be identified as such in the Methods section of the paper, with rationale. If a reported analysis was performed based on an interesting but unanticipated pattern in the data, please be clear that the analysis was data-driven.

Please ensure that the study is reported according to the STROBE guideline, and include the completed STROBE checklist as Supporting Information. Please add the following statement, or similar, to the Methods: "This study is reported as per the Strengthening the Reporting of Observational Studies in Epidemiology (STROBE) guideline (S1 Checklist)." The STROBE guideline can be found here: http://www.equator-network.org/reporting-guidelines/strobe/ When completing the checklist, please use section and paragraph numbers, rather than page numbers which will likely no longer correspond to the appropriate sections after copy-editing.

Results: 

Please incorporate supplementary table A into the main paper as a table showing the baseline characteristics of the study population (to become Table 1) which should include the information included in paragraph 1 of your Methods section.

Please report the number of participants early in your Results section.

For the ORs reported, please specify the comparison groups / reference point.

For the observational component of your study, please provide the actual numbers of events for the outcomes in a table, not just ORs.

Line 276 - please revise to ‘casually associated with’

Line 325 - please clarify what is meant by ‘different effects’

Discussion:

Lines 368 / 448 / 464 - Please temper assertions of primacy by adding ‘to the best of our knowledge’ or similar

Figures:

Please consider avoiding the use of red and green in order to make your figure more accessible to those with colour blindness.

Please define all abbreviations used in the figure legend of each figure.

Figure 1 - please clarify whether there is a part B missing

Figure 4 - please revise to ‘estimated effect(s)’ or similar

Figure 5a - please label the axes

Figure S1 - please use bolder colours in your plots as the bars and whiskers are difficult to see. 

Tables:

Please define all abbreviations used in the table legend of each table, including supplementary tables.

When a p value is given, please specify the statistical test used to determine it in the table legend.

Table 2 - should the final column read P value (since most rows are adjusted for BMI / WHR / WHRadjBMI)?

Supplementary files: 

Please separate supplementary tables into S1 table, S2 table etc rather than S1 Table A / B etc. so that they can be individually accessed.

Please provide titles and legends for each individual table and figure in the Supporting Information.

Comments from the reviewers:

Reviewer #1: 

This is a very important work comprehensively examining the role of obesity in multiple female reproductive conditions. Observational estimates as well as genetically predicted effects of BMI and central adiposity were assessed. The Mendelian randomization techniques used represent the latest developments in the field. Complex patterns of causality between adiposity traits and reproductive conditions were observed. The Discussion accurately presents the findings with a view to future directions. Clarification of a few details will assist readers: 

1. Participants in UKBB were age 40-69 at recruitment. Many of the conditions examined present at younger ages and may even resolve at older age (PCOS, for example). The conditions are identified based on ICD9 and ICD10 codes. If a condition happened in the past (or resolved in the past), there is a good chance that it would not be recorded in ICD codes. This may explain the extremely low rate of PCOS (0.3%). This limitation should be discussed in more detail, as it may explain the low rates of several conditions in addition to underdiagnosis. 

2. Line 123: P < 5E-9 is quoted. However, 5E-8 is much more commonly used as the genome-wide significance level. Please correct or explain. 

3. Line 127: "waist-specific and hip-specific WHR" is unclear. What was meant does not become clear until reading results several pages later. 

4. On line 99, the UKBB numbers for PCOS are given as 746 cases and 256,447 controls. However, line 151-152 describes a UKBB GWAS of 11,186 cases and 273,812 controls. Please reconcile the numbers. 

5. As there are multiple methods of calculating F statistic, the method used here should be given. 

6. The Walford GWAS for Stumvoll insulin sensitivity index found very few signals at genome-wide significance. What are the implications of that for the current study? 

7. Figure 1 highlights instances where "non-linear models fit the data better than linear models" whereas line 289 states that "no non-linear models explained the causal effects of BMI … better than linear MR models." Please clarify. 

8. Effects of three potential mediators (leptin, FI, IR) were examined individually. Is there an opportunity here to examine them jointly?

9. "junk cluster" seems to apply a value judgement to certain variants. However, if that is the accepted term for this technique, it can be used. 

10. Metformin should not be referred to as "first line treatment' (line 486). 

Reviewer #2: The authors present a very interesting and detailed manuscript exploring the role of obesity-related traits in relation to a number of reproductive and menstrual conditions. They perform both observational and Mendelian randomization analysis, including the use of bi-directional MR and two-step approaches to investigate mediating mechanisms. Further, they look at the use of genetic variants to provide insights into the clustering of reproductive conditions in relation to adiposity. 

1. In the Introduction, more specific details from previous studies assessing the relationship between obesity and reproductive traits would be useful, including effect estimates and the relative strengths and limitations of these previous approaches in order to highlight motivation for the present analysis. 

2. After the statement "Women with advance-stage endometriosis have lower BMI than those with minimal disease, and the inverse BMI-endometriosis association is stronger in women with infertility" , the authors should specifically highlight potential issues of reverse causation and selection bias here. 

3. In the Methods, there are issues with using the GWAS for WHRadjBMI, which may introduce bias into the MR analysis: http://dx.doi.org/10.1093/ije/dyaa266 . Instead, could the authors perform MVMR with both WHR and BMI in same model? 

4. There are issue of temporality with the adiposity measures often assessed after the diagnosis of reproductive conditions in the UK Biobank. While this is likely to be less of a thread in the MR analysis, this does have implications for the observational analysis. Could the authors run a sensitivity analysis using reported adiposity at an earlier timepoint? For example, in UK Biobank the participants were asked about their comparative body size at age 10. 

5. The authors mention weighting the GRS but there are issues of introducing bias due to winner's curse and sample overlap when doing this (as mentioned). Would it be possible to run an MR sensitivity analysis where the SNPs are weighted by the effects seen in FinnGen? 

6. The authors present power calculations but it is unclear how the results of a-priori power calculation were used to inform on whether further analyses should be conducted. 

7. The authors mention that the IVW method was selected as the preferred method based on the Rucker framework but they should still acknowledge susceptibility of this test to horizontal pleiotropy 

8. Did the authors consider the use of Steiger filtering when assessing the bi-directional relationships? 

9. The authors should consider comparing the results of the two-step MR product of co-efficients estimates with those obtained from MVMR and difference method for estimating mediation. 

10. More details on the use of UMAP for SNP clustering would be useful. 

11. Could the authors provide more information on the distinction between the non-linear observational models (generalized additive models and fractional polynomials) and how these approaches compare with the non-linear MR? 

12. How were AICs compared to determine best fit of non-linear models? Was a test for difference in AIC performed? 

13. In Table 1, the estimates for miscarriage (multiple consecutive) are missing for the observational analysis. Include a footnote to indicate why this is the case. 

14. Did the authors perform a test for difference to compare the observational and MR aresults? 

15. The authors report that MR estimates of association between BMI and HMB, endometriosis and PCOS were attenuated compared with observational. However, for both HMB and PCOS there is a small but very precise effect from the MR (e.g. OR 1.01 for HMB). It is unusually to get such a precise MR effect and CIs appear to be smaller here than the corresponding observational analysis. Could the authors explain this? 

16. The authors should compare the IVW results with those from more pleiotropy-robust methods (shown in from Figure S1) in the text 

17. The authors claim that non-linear models explained the observational data better than linear models, but this was not the case for the MR analysis. As well as indicating potential low power in the MR to detect non-linear relationships, could the authors speculate on any other reasons for this? 

18. Unless I missed it, I was unclear why non-linear MR was only run for UF, HMB and stillbirth/miscarriage. Was this indicated based on power calculations? 

19. Unless I missed it, I was unclear why reverse MR was only run for endometriosis, PCOS and UF? Was this indicated based on power calculations? 

20. Could the authors consider the use of weak instrument correction methods for the reverse MR? 

21. The authors perform two-step analysis considering leptin, fasting insulin and insulin sensitivity as mediators. Was there any evidence that these factors in turn influence adiposity (i.e. may also be potential confounders?) This will have implications for the estimation of the mediated effect. 

22. It is interesting that leptin, fasting insulin and ISI were associated with risk of PE after adjustment for BMI, but not in unadjusted analysis. Why might this be? 

23. The authors should explain the likely reason for the high proportion of women reporting "unsure" for menopause - e.g. having had a hysterectomy. 

24. There is a detailed discussion contextualizing findings but more about the relative strengths/limitations of MR (including assumptions) and other genetic approaches used in this context would be useful. 

Reviewer #3: I would like to congratulate the authors on a very comprehensive and convincing study on the effect of obesity on female reproductive conditions. Venkatesh et al use both

observational and genetic study design to triangulate evidence and use a wide range of methodology ranging from logistic regression to Mendelian randomization (MR), and also include advanced methods like for example nonlinear models and mediation analysis. Also I would like to thank the authors for providing the R-code as a github repository; this is much appreciated and a great example for reproducible science. 

My comments are mainly focused on the wording, in particular the careless use of the "causal" word, and the results from the mediation analysis.

Major comments:

1. Interpretation of MR estimate: The authors interpret the MR effect estimate throughout the manuscript as "causal". MR is a causal inference method, but MR relies on strong and mainly untestable assumptions. Consequently, I would urge the authors to adapt a more cautious interpretation of the MR effect estimate in the Abstract, Methods, and Results section in line with the MR guidelines [1]. Please see a nuanced discussion how MR estimates should be interpreted in Section 9 of the guidelines [1].

For example, instead of distinguishing between observational and "causal" associations, the authors may distinguish between observational and "genetic" associations. Instead of "Causal effect estimates" the authors may use the term "MR effect estimates". Finally, phrases like "Genetically predicted visceral adipose tissue mass was causal for" should be changed into "Genetically predicted visceral adipose tissue mass was associated with". 

Following the MR guidelines [1] the authors could then add a paragraph discussing the "causal interpretation" in the light of the plausibility of the instrumental variable assumptions, the concordance of estimates from different methods and different analytical approaches, the results from sensitivity and supplementary analyses. Here, the authors can make a very strong point given the extensive analysis presented in the manuscript that these findings are indeed likely to be causal.

2. Mediation analysis: The results of the multivariable MR (Table 2) describing the effect of leptin, fasting insulin and insulin sensitivity on pre-eclampsia are surprising. Could the authors please comment on why there is no significant effect in the unadjusted univariable MR model (which instruments are used here?) and there is a highly significant effect in a multivariable MR model when adjusting for BMI (are the same instruments used as in the univariable model or does this analysis include BMI instruments as well)? When interpreting multivariable MR results in terms of total and direct effect [2], how can the total effect of an exposure be Null, while there is a strong direct effect? Is there a biological explanation for this? 

3. Ruecker's framework: To the best of my knowledge Ruecker's framework (please note Rücker or Ruecker, not Rucker) can be used to distinguish if the IVW or MR-Egger method provides the better model fit. It cannot be used to compare results from the weighted median MR, because the Q-statistic is not defined in the median MR model. Please correct on page 8, line 163 and remove weighted median from your list. No changes need to be made to the results or Table F in the Supplementary. It would be perfectly appropriate to consider IVW being the most powerful method as the main method and MR-Egger and the weighted median as sensitivity analysis providing more pleiotropy-robust inference. 

Minor comments:

- FDR adjustment: When performing FDR correction, for how many tests did you adjust? The number of exposures, the number of outcomes, or the number of exposures times the number of outcomes? Please clarify. Which method did you use for FDR correction? Please provide a citation.

- Page 6, line 123, genome-wide significance, why did you use 5E-9 and not 5E-8?

- Page 11, line 223, please introduce the abbreviation UMAP.

- Page 11, line 227, MR-Clust v0.1.0 R package (56). Is citation (56) to Pers et al correct? Or should this rather be citation (57) Foley et al?

References

1. Burgess, S. et al. Guidelines for performing Mendelian randomization investigations [version 2; peer review: 2 approved]. Wellcome Open Res. 4, (2020).

2. Burgess, S., Daniel, R. M., Butterworth, A. S., Thompson, S. G. & the EPIC-InterAct Consortium. Network Mendelian randomization: using genetic variants as instrumental variables to investigate mediation in causal pathways. Int. J. Epidemiol. 44, 484-495 (2015).

[LINK]

---

## [Decision Letter · Decision Letter 2]

17 Nov 2021

Dear Dr. Venkatesh,

Thank you very much for re-submitting your manuscript "Obesity and risk of female reproductive conditions: A Mendelian Randomisation Study" (PMEDICINE-D-21-02467R2) for review by PLOS Medicine.

I have discussed the paper with my colleagues and the academic editor and it was also seen again by three reviewers. I am pleased to say that provided the remaining editorial and production issues are dealt with we are planning to accept the paper for publication in the journal.

[LINK]

We look forward to receiving the revised manuscript by Nov 24 2021 11:59PM.   

Sincerely,

Caitlin Moyer, PhD

Associate Editor

PLOS Medicine

on behalf of,

Louise Gaynor-Brook, MBBS PhD

Associate Editor 

PLOS Medicine

plosmedicine.org

Requests from Editors:

1. Data availability statement: Please include the sentence from the Methods at line 273-274 with the link for the analysis scripts. “All scripts used in analyses are deposited at https://github.com/lindgrengroup/obesity_femrepr_MR”

2. Title: Please change to “A Mendelian randomization study” as the subtitle.

3. Abstract: Lines 12-18: Please quantify the main results presented in the Abstract with ranges of ORs, 95% CIs and p values for the BMI-related measures associated with each reproductive disorder.

4. Abstract: Line 25: Please avoid the use of language that implies causality (e.g. please replace “increased risk of” with “were associated with increased risk of”).

5. Abstract: Line 26: We suggest removing “one of the first” from the sentence.

6. Author summary: Line 43: Please revise to: “We found that inherited genetic variation that is associated with obesity…”

7. Introduction: Line 59: Please clarify the sentence to indicate the direction of the relationship (for example, is increased BMI associated with increased prevalence of gynaecological conditions?)

8. Methods: Line 141-142: Prospective analysis plan. Thank you for noting that no prospective analysis plan was followed. Please state in the Methods section when each of the analyses were planned, and if/when reported analyses differed from those that were planned. For example, any changes made in the analyses, including those made in response to peer review comments, should be identified as such in

the Methods section of the paper, with rationale. If a reported analysis was performed based on an interesting but unanticipated pattern in the data, please be clear that the analysis was data-driven.

9. Methods: STROBE statement: Thank you for including the STROBE Checklist. We request that you please also include and report your study according to the recently published STROBE-MR checklist (Skrivankova VW, Richmond RC, Woolf BAR, Yarmolinsky J, Davies NM, Swanson SA, VanderWeele TJ, Higgins JPT, Timpson NJ, Dimou N, Langenberg C, Golub RM, Loder EW, Gallo V, Tybjaerg-Hansen A, Davey Smith G, Egger M, Richards JB. Strengthening the Reporting of Observational Studies in Epidemiology Using Mendelian Randomization: The STROBE-MR Statement. JAMA. 2021;326(16):1614-1621.)

Please add the following statement, or similar, to the Methods: "This study is reported as per the Strengthening the Reporting of Observational Studies in Epidemiology Using Mendelian Randomization (STROBE-MR) guideline (S1 Checklist)."

10. Results: Line 280: Please report p values as p<0.001, unless there is a rationale for presenting the exact p value. Please also provide 95% CIs (here and throughout where p values are presented).

11. Results: Line 281-283: Please report both the 95% CIs and p values for these associations.

12. Results: Line 294: Please clarify in the Methods if this term was used when participants were surveyed. We suggest revising “comparatively plumper than average” to something that might be more universally clear such as “having comparatively greater adiposity than average” or similar, here and in S6 table.

13. Results: Line 294-303: Please report both the 95% CIs and p values for these associations.

14. Results: Line 312-313: Please revise to “While genetically predicted BMI was also associated with increased risk of most…”

15. Results: Line 319-320: Please revise to “The differential association of genetically predicted body fat distribution with female reproductive traits…”

16. Results: Line 347: Please revise the reference call-out style here to [35,36].

17. Results: Line 392-394: Please revise “effect on” to “associations with” in this sentence.

18. Discussion: Line 419: Please avoid “one of the first” or temper statements regarding primacy with “to the best of our knowledge…” or similar.

19. Discussion: Line 420: Please avoid the use of language that implies causality (e.g. please replace “increase risk of” with “ associated with increased risk of”).

20. Discussion: Line 421: Please clarify what is meant by “whose effects may be non-uniform” in the sentence. We suggest revising to “...a broad range of reproductive conditions, and these associations may be non-uniform across the obesity spectrum.” or similar.

21. Discussion: Line 422-432: Please replace “effects” with “associations” in this sentence.

22. Discussion: Line 439: Please revise to: “with each S.D. increase in BMI associated with double the risk of preeclampsia...:”

23. Discussion: Lines 452-453: Please revise to: “...the MR analyses indicated that each S.D. higher genetically predicted BMI was associated with an increased PCOS risk of only 13%. Similarly, 1 S.D. higher genetically predicted WHR and WHRadjBMI was associated with a 24% increase in endometriosis risk, while the observational analyses suggest associations with more modest increases in risk of 7% and 2% respectively.” or similar.

24. Discussion: Lines 488-489: Please revise to: “Genetically predicted visceral adipose tissue mass was associated with iincreased risk of PCOS and pre-eclampsia…”

25. Discussion: Line 502: We suggest replacing “plumper” with “having greater than average adiposity” or similar.

26. Discussion: Line 523-524: We suggest revising to: “Finally, genetic clustering of female reproductive conditions revealed evidence supporting common genetic causes of obesity on endometriosis, UF, and HMB, which are known to share mechanisms of development.”

27. Discussion: Line 550: Please change “on participants” to “with participants” in this sentence.

28. Discussion: Line 557: We suggest changing to “...obesity is associated with increased risk of female reproductive disease” or similar.

29. Discussion: Line 558: We suggest changing “cheap” to “affordable” in this sentence.

30. References: In reference 29 and 47, please remove the disclosure information.

31. Table 2,Table 3, and Table 4: Please present p values as p<0.001 where applicable, unless there is rationale to present exact p values.

32. Table 2: Please note factors adjusted for in the observational analyses in the legend. Please also provide results from unadjusted analyses.

33. Figure 5: In panel A, please define the “x” and “y” axes.

34. Supporting Information Figures: We suggest formatting all parts of each figure/caption to fit together within 1 file.

35. S16 Table: There is an extra “hormones.” at the end of the Table title in the file.

36. Supporting information Tables (for example, S12 Table): Please report p values as p<0.001 where applicable, except under circumstances where there is rationale for reporting the exact p values.

Comments from Reviewers:

Reviewer #1: The critiques have been well addressed. 

Reviewer #2: The authors have adequately responded to my previous comments and I commend them on an excellent paper. 

Reviewer #3: I thank the Authors for addressing my comments.

[LINK]

---

## [Editor Report · Decision Letter 3]

7 Dec 2021

Dear Dr Venkatesh, 

On behalf of my colleagues and the Academic Editor, Sarah Stock, I am pleased to inform you that we have agreed to publish your manuscript "Obesity and risk of female reproductive conditions: A Mendelian randomization study" (PMEDICINE-D-21-02467R3) in PLOS Medicine.

Additionally, please address the following editorial points:

1. Figure 5 panel A: Thank you for providing the explanation, please display the UMAP plots according to the standard practice in the field.

2. Formatting of S1 and S2 figure panels: Please combine separate parts (e.g., S1A and S1B Table) belonging to the same figure into one file (the panels do not need to all fit together on the same page).

3. Supporting Information: Please do not include a “supplementary captions” file. Please list all Supporting Information item titles and legends at the end of the main text of the manuscript.

4. STROBE-MR checklist: Please provide a revised version of the checklist using section and paragraph numbers to refer to locations in the text (e.g. Methods, paragraph 1). Please do not use page or line numbers. For the sections including “Other Information” please refer to Competing Interests, Funding, and Data Availability sections.

PRESS

Sincerely, 

Caitlin Moyer, PhD

Associate Editor

on behalf of,

Louise Gaynor-Brook, MBBS PhD 

Senior Editor 

PLOS Medicine